# From Talking to Singing: A New Challenge for Audio-Visual Deepfake Detection

**Ke Liu** [1]   **Jiwei Wei** [1]   **Wenyu Zhang** [1]   **Shuchang Zhou** [1]   **Ruikun Chai** [1]   **Yutao Dai** [1]   **Chaoning Zhang** [1]
**Yang Yang** [1]

## Abstract

With rapid advances in audio-visual generative models, reliable forgery detection becomes increasingly critical. Existing methods for audio-visual deepfake detection typically rely on cross-modal inconsistencies. In singing, rhythmic vocalization weakens this coupling and introduces a nontrivial domain shift, substantially degrading detection performance. We construct the **S**inging **H**ead **D**eep**F**ake (SHDF) dataset using rhythm-aware generative models to fill the gap in singing benchmarks. To cope with cross-scenario domain shifts, we propose a **T**ext-guided **A**udio-**V**isual **F**orgery **D**etection (T-AVFD) framework that generalizes across both talking and singing scenarios. T-AVFD comprises a facial authenticity pattern learner and a multi-modal differential weight learning module. The pattern learner aligns facial features with multigranularity textual descriptions to learn generalizable authenticity patterns. The weight learning module preserves intrinsic audio-visual consistency and adaptively integrates it with authenticity patterns via differential weighting. Extensive experiments on multiple talking head deepfake datasets and SHDF show consistent improvements over existing baselines and strong robustness under diverse perturbations. The project page is available at https://LiuKe3068LikWix.github.io/SingingHead-DeepFake/.

## 1. Introduction

The rapid advancement of generative AI has enabled individuals without extensive technical expertise to effortlessly create high-quality audio-visual content (Peng et al., 2024;

[1]Center for Future Media, School of Computer Science and Engineering, University of Electronic Science and Technology of China, Chengdu, China. Correspondence to: Jiwei Wei <mathematic6@gmail.com>.

*Proceedings of the 43rd International Conference on Machine Learning*, Seoul, South Korea. PMLR 306, 2026. Copyright 2026 by the author(s).

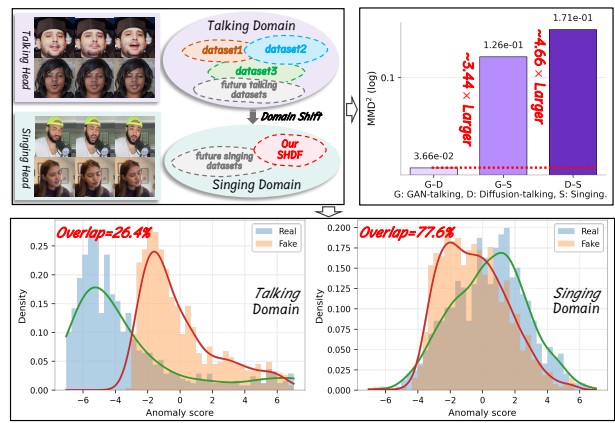

*Figure 1.* **Domain shift diagnostics.** Compared to talking, singing induces a nontrivial domain shift for audio-visual forgery detection. Using the forgery-agnostic AVH-Align detector (Smeu et al., 2025), we quantify cross-domain discrepancy with $MMD^2$ (top-right) and examine anomaly score separability via the real-fake distribution overlap (bottom). Singing exhibits substantially larger shift from talking domains (G-S and D-S are $\sim3.44\times$ and $\sim4.66\times$ larger than G-D in $MMD^2$), while its score distributions show much heavier real-fake overlap (77.6% vs 26.4%), indicating reduced separability under this shift.

Su et al., 2024). While this progress benefits creative expression and digital entertainment (Ye et al., 2023; Liu et al., 2025a), it also raises serious concerns due to the potential misuse of synthetic media (Yan et al., 2025). Consequently, audio-visual forgery detection has received growing attention (Smeu et al., 2025). Despite notable progress, current detectors remain brittle under domain shifts.

Traditional forgery detectors often rely on unimodal manipulation traces in either the facial or audio stream (Liu et al., 2024a; Huang et al., 2025), which become less reliable as advanced audio-visual generators improve cross-modal realism. Recent research has moved toward multi-modal detection, primarily identifying forgeries through audio-visual inconsistencies (Liu et al., 2024b; Smeu et al., 2025). However, most methods are still developed for talking scenarios, while singing remains underexplored despite introducing a nontrivial domain shift from talking.

To make this gap concrete, we perform domain-shift diagnostics from talking to singing using the squared Maximum

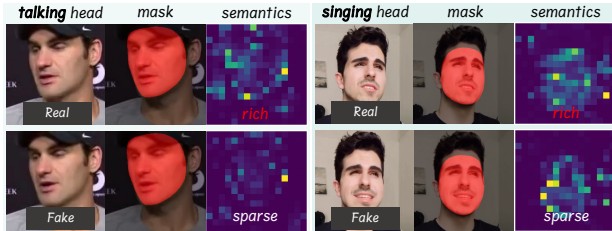

*Figure 2.* **Facial semantic pattern visualization.** In both singing and talking scenarios, real samples display richer facial semantics, while fake samples appear more sparse.

Mean Discrepancy ($MMD^2$) and assess separability via score-distribution overlap. Figure 1 shows that the talking-to-singing discrepancy is substantially larger in $MMD^2$. Moreover, singing exhibits markedly heavier real–fake overlap in detector scores, indicating reduced separability and challenging cross-scenario transfer. To enable controlled evaluation under this singing-induced shift, we construct the Singing Head DeepFake (SHDF) dataset using rhythm-aware generative models, including MEMO (Zheng et al., 2024), Hallo2 (Cui et al., 2025a), and EchoMimic (Chen et al., 2025). SHDF contains real and synthetic singing videos across diverse identities and genders. Compared to talking, singing forms a distinct audio-visual regime in which speech-driven alignment cues can be less stable, motivating detectors that go beyond conventional audio-visual alignment evidence.

Existing methods demonstrate promising performance in talking head deepfakes by evaluating the consistency between lip movements and speech. However, in singing scenarios, musical accompaniment and rhythmic vocalization can reduce the reliability of alignment-centric evidence, motivating complementary cues that transfer across scenarios. To this end, we propose a Text-guided Audio-Visual Forgery Detection (T-AVFD) framework that leverages facial semantic information as an auxiliary authenticity signal. As shown in Figure 2, both singing and talking heads exhibit significantly richer and more coherent facial semantic representations in real samples than in synthetic ones. Such semantics, not solely tied to lip–speech alignment, provide discriminative information for generalized audio-visual forgery detection.

To avoid overfitting to generator-specific forgery signatures, T-AVFD is trained exclusively on real talking samples. This choice aligns training with established talking-centric baselines while reserving singing as a strictly unseen target domain for evaluating cross-scenario generalization. T-AVFD comprises two components: a Facial Authenticity Pattern Learner (FAPL) and a Multi-Modal Differential Weight Learning (MMDWL) module. FAPL aligns facial representations extracted by Alpha-CLIP (Sun et al., 2024) with multi-granularity text descriptions in a face–text contrastive

space, yielding scenario-agnostic authenticity patterns that remain discriminative under the talking-to-singing shift. While lip–speech consistency has been the canonical cross-modal cue for audio-visual forgery detection (Smeu et al., 2025), its reliability can degrade markedly under domain shifts (Figure 1). MMDWL therefore preserves alignment-related representations via a pre-trained lip-reading expert (Shi et al., 2022) and complements them with the learned facial authenticity patterns. Under deepfake scenario changes, the relative reliability of these two signals can vary. Accordingly, MMDWL predicts content-conditioned adaptive weights and applies a modulation bias to stabilize fusion under cross-scenario transfer.

In summary, our main contributions are as follows:

- We propose a Singing Head DeepFake dataset that, for the first time, extends audio-visual forgery detection from talking to singing scenarios. This introduces a novel and highly challenging benchmark for evaluating the performance of detection models.

- We identify generalized facial authenticity patterns as a key to robust fake detection. Building on this insight, we propose a unified audio-visual forgery detection framework, T-AVFD, that jointly models facial semantics and lip-speech consistency.

- Extensive experimental results show that our model exhibits strong generalization when facing forgery attacks in both singing and talking scenarios, significantly outperforming baseline methods.

## 2. Related Work

### 2.1. Audio-Visual Datasets for Deepfake Detection

Early deepfake datasets mainly rely on face-swapping techniques (Rosberg et al., 2023), which introduce visible artifacts such as boundary inconsistencies and are relatively easy to detect. As talking head generation becomes prevalent, the focus shifts to manipulating lip movements and facial expressions while preserving identity, making forgery detection more challenging. Audio-visual datasets like FakeAVCeleb (Khalid et al., 2021) and FaceForensics++ (Rossler et al., 2019) typically include fake videos generated by GAN-based models (Prajwal et al., 2020) or partial facial manipulation methods (Thies et al., 2016). Recent datasets such as AVLips (Liu et al., 2024b), LAV-DF (Cai et al., 2023), AV-DeepFake1M (Cai et al., 2024), and PolyGlobFake (Hou et al., 2024) adopt talking head generation with strong lip-sync performance, which increases the complexity of existing detection frameworks. TalkingHeadBench (Xiong et al., 2025) further advances head realism and diversity by introducing diffusion-based generation, setting a higher bar for evaluating detection methods.

However, these datasets are predominantly centered on talking scenarios, leaving the talking-to-singing shift largely unexamined. To quantify this shift and enable controlled cross-scenario evaluation, we present SHDF, a singing-focused dataset for audio-visual forgery detection.

## 2.2. Audio-Visual Deepfake Detection

Audio-visual head generation involves translating acoustic signals into temporally coherent facial motions, while capturing cross-modal dependencies between audio and visual streams (Wei et al., 2020; 2023; Chen et al., 2024; Cao et al., 2025; Liu et al., 2025b). The increasing sophistication of talking head generation has substantially raised the difficulty of forgery detection, prompting a shift from unimodal feature modeling to joint learning across audio and visual streams (Zheng et al., 2021; Liu et al., 2024b). Most detection methods operate under supervised learning paradigms, either by training on labeled real-fake videos (Chugh et al., 2020; Mittal et al., 2020), or by leveraging self-supervised pretraining followed by fine-tuning with annotated data (Haliassos et al., 2022). However, the reliance on labeled samples limits generalization and robustness. To mitigate this, recent detectors explore unsupervised alternatives that utilize only authentic data for training (Li et al., 2024; Ricker et al., 2024). For example, AVAD (Feng et al., 2023) adopts an anomaly detection framework based on autoregressive modeling of audio-visual synchronization. AVH-Align (Smeu et al., 2025) further enhances resilience by learning self-supervised representations and reducing sensitivity to dataset-specific artifacts.

Although these methods learn effective detection patterns, their reliance on alignment-centric evidence limits transferability under the talking-to-singing domain shift. In contrast, we propose the T-AVFD framework based on general visual authenticity patterns, enabling effective generalization across both talking and singing forgeries.

## 3. Singing Head DeepFake Dataset

To the best of our knowledge, most publicly available audio-visual deepfake datasets are synthesized using talking speech, and none provide samples specifically designed for detecting singing head forgeries. To address this gap, we first collect real singing videos from YouTube, covering 80 distinct identities. Based on this real data, we further synthesize 3,000 audio-visual samples using advanced generative models conditioned on rhythm. To enhance dataset diversity, we incorporate 20 additional identities from HDTF (Zhang et al., 2021) during the generation process.

The overall construction pipeline of our Singing Head Deep-Fake (SHDF) dataset is illustrated in Figure 3. Considering both the feasibility and the responsiveness to song-driven

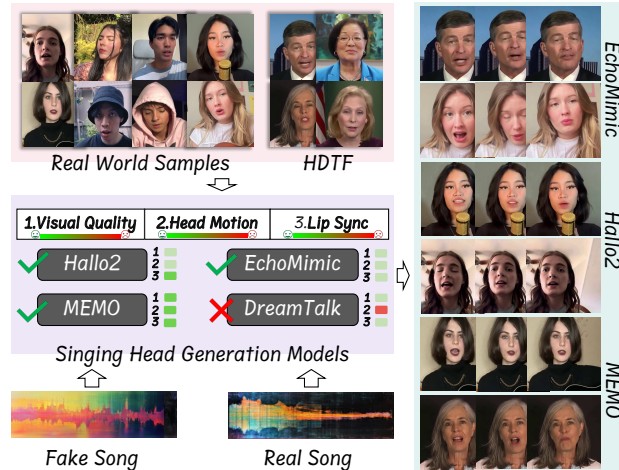

*Figure 3.* **SHDF dataset construction.** We generate high-quality videos using audio-visual synthesis methods guided by musical rhythm and vocal prosody, resulting in expressive facial movements, natural head motion, and accurate lip synchronization.

head poses and facial expressions, we select MEMO (Zheng et al., 2024), Hallo2 (Cui et al., 2025a), EchoMimic (Chen et al., 2025), and DreamTalk (Ma et al., 2023) as candidate models. To ensure the quality of synthesized data, we conduct an evaluation from three aspects: visual quality, head motion, and lip-sync accuracy. Under singing-driven scenarios, MEMO demonstrates consistently strong performance across all three dimensions. Hallo2 performs well in lip-sync accuracy, with satisfactory performance in visual quality and head motion. EchoMimic performs at a usable level across all aspects. In contrast, DreamTalk suffers from rigid head poses. Therefore, we select the first three models as the foundation for forgery sample generation. For the driving audio, we use real songs collected from YouTube and fake songs synthesized by Sonics (Rahman et al., 2025). Based on the performance differences among synthesis models, we adopt MEMO as the primary synthesis model to generate 2,000 samples, while EchoMimic and Hallo2 each contribute 500 samples. Overall, SHDF comprises 2,600 real samples from 80 distinct individuals and 3,000 forged samples synthesized from 100 identities.

## 4. Method

In Figure 4, given video frames $\{F_t\}_{t=0}^{T}$, mouth crops $\{M_t\}_{t=0}^{T}$, and audio $\{A_t\}_{t=0}^{T}$, T-AVFD outputs a video-level detection score $s$. T-AVFD first learns an authenticity pattern $fp$ by aligning face semantic $f$ with multi-granularity positive prompts $\{p_i\}_{i=0}^{g}$ while repelling it from negative prompts $\{n_i\}_{i=0}^{g}$ via $\mathcal{L}_{ft}$. In parallel, a lip-reading model encodes $\{M_t\}_{t=0}^{T}$ and $\{A_t\}_{t=0}^{T}$ into audio-visual alignment features $(v, a)$. $fp$ and $(v, a)$ are then adaptively weighted and fused to produce the final score $s$. Training

*Table 1.* **Predefined text prompts.** We define positive-negative descriptions at multiple facial granularities.

| Region | Positive Description | Negative Description |
|--------|---------------------|---------------------|
| Face | *a real human face* | *a fake human face* |
| Eyes | *a bonafide face with expressive eyes* | *a spoof face with dull eyes* |
| Mouth | *a genuine face with natural mouth* | *a forged face with unnatural mouth* |

uses only real talking videos and optimizes $\mathcal{L} = \mathcal{L}_{ft} + \mathcal{L}_{av}$ without any synthetic samples.

### 4.1. Facial Authenticity Pattern Learner

**Face Semantic Extraction.** Motivated by the significant semantic gap between real and fake samples observed in Figure 2, we construct a hybrid input $\{F_t\}_{t=0}^{T}$ by combining video frames with their corresponding facial masks, and feed it into Alpha-CLIP (Sun et al., 2024) to learn semantic features in the facial region. Here $T$ denotes the number of video frames. As shown in Figure 4, Alpha-CLIP is a variant of CLIP (Radford et al., 2021) that augments the original RGB image encoder $E_f$ with an additional mask encoder $E_m$ to specify regions of interest. Leveraging transformer-based attention mechanisms $AT_{fm}$, it enables region-level semantic learning. Unlike traditional cropping or masking methods, Alpha-CLIP preserves global image context while enhancing regional semantic understanding, making it an ideal face encoder for our T-AVFD. To obtain a stable face semantic $f$, face features extracted from all frames are averaged to capture consistent semantic patterns while reducing temporal fluctuations and content-specific biases.

**Multi-Granularity Positive and Negative Text Features.** To obtain multi-granular positive and negative text pairs that describe facial attributes, we assign positive texts $\{p_i\}_{i=0}^{g}$ to real faces and use negative texts $\{n_i\}_{i=0}^{g}$ for contrastive supervision in the face-text alignment space, where $g$ denotes the granularity level. We use ChatGPT (Achiam et al., 2023) to generate all positive and negative texts. The text pairs are constructed from three granularity levels: face, eyes, and mouth. Table 1 presents the predefined positive-negative region descriptions used in our prompt hierarchy. Table 4 in the experimental section validates the effectiveness of our text prompt design.

The text embedding space of CLIP is primarily designed for general visual concepts. To enable the CLIP text encoder to capture semantics related to both real and fake content, we incorporate learnable tokens into fixed text pairs. Given the negative text $n_i$ and the positive text $p_i$, we introduce $l$ learnable tokens and concatenate them with each tokenized text embedding. The resulting sequences are encoded by the CLIP text encoder to produce intermediate representa-

tions $f_i^p$ and $f_i^n$. We average and normalize them to obtain polarity-opposed text features, which are then projected through a shared linear mapping to produce $p$ and $n$:

$$p = W\left(\frac{1}{g_p}\sum_i \frac{f_i^p}{\|f_i^p\|}\right), n = W\left(\frac{1}{g_n}\sum_i \frac{f_i^n}{\|f_i^n\|}\right), \quad (1)$$

where $W$ denotes learnable weights, and $g_p$, $g_n$ represent the granularity levels of positive and negative text prompts, both set to 3. The shared-weight linear layer prevents $n$ and $p$ from being projected into separate feature subspaces, ensuring consistent semantic mapping.

**Face-Text Contrastive Alignment.** A key challenge in unsupervised audio-visual forgery detection lies in learning forgery-aware representations without relying on synthetic samples. To this end, we design a Face-Text Contrastive Alignment (FTCA) loss $\mathcal{L}_{ft}$ that leverages textual guidance to distill discriminative visual cues from real faces. Formally, given the facial feature $f$, the positive textual embedding $p$ describing the authentic face attributes, and the negative textual embedding $n$ representing inconsistent semantics, we align $f$ with $p$ while pushing it away from $n$. Each feature is normalized before computing cosine similarity:

$$s^+ = \frac{f^\top p}{\tau}, s^- = \frac{f^\top n}{\tau}, \quad (2)$$

where $\tau$ denotes the temperature parameter controlling the contrast sharpness. The loss is formulated as a binary contrastive objective:

$$\mathcal{L}_{ft} = -\frac{1}{N}\sum_{i=1}^{N}\log\frac{\exp(s_i^+)}{\exp(s_i^+) + \exp(s_i^-)}, \quad (3)$$

where $N$ is the number of visual-text pairs. Trained on real data, FTCA captures authentic facial patterns. When such patterns are violated in forged videos, the resulting distributional deviations serve as cues for generalizing to various forgery types. To enhance semantic consistency, we concatenate $p$ with $f$ to obtain the target authentic facial patterns $fp$.

### 4.2. Multi-Modal Differential Weight Learning

**Audio-Visual Feature Extraction.** Despite the limitations of audio-visual consistency in complex forgery scenarios, the inherent dual-modality structure of talking and singing videos makes cross-modal consistency (Wei et al., 2021) a fundamental basis for forgery detection. Therefore, we adopt a self-supervised audio-visual representation learning model (Shi et al., 2022) to extract visual and audio features. This lip-reading model is pre-trained on large-scale talking head videos (Son Chung et al., 2017), learning temporally synchronized and semantically consistent audio-visual representations that capture cross-modal correspondences at the phoneme level.

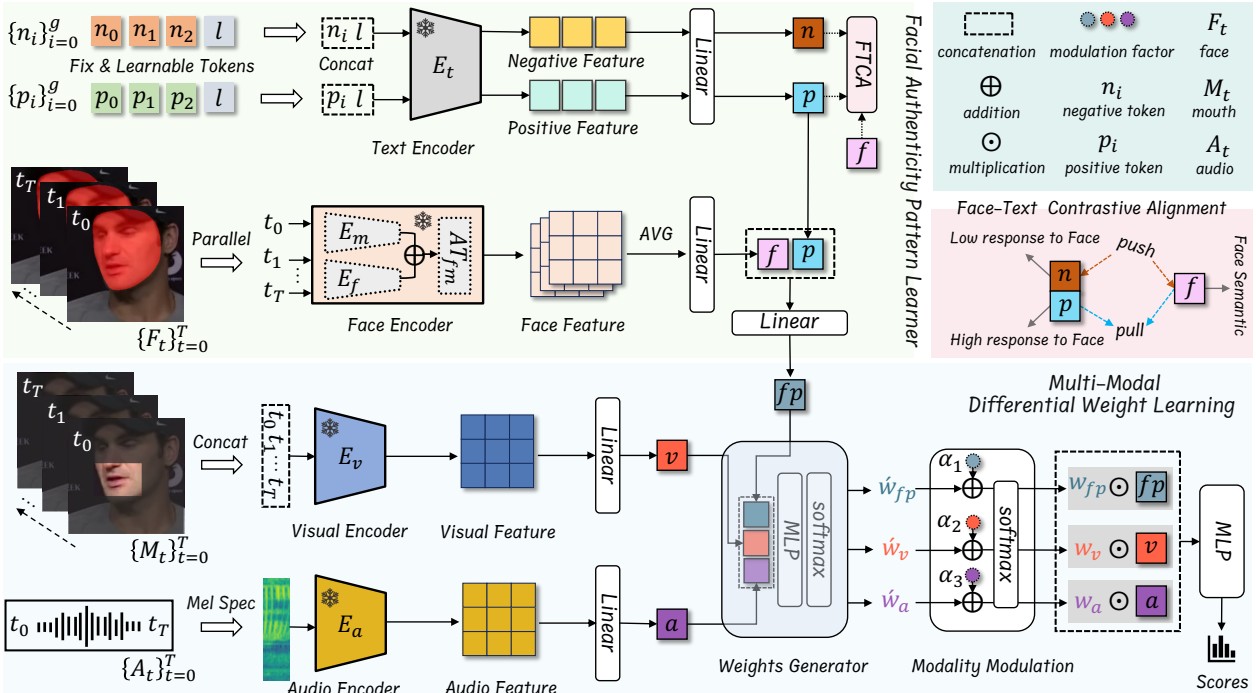

*Figure 4.* **Overview of the proposed T-AVFD framework.** The face encoder is employed to extract semantic features from the facial region. Multi-granular text prompts, augmented with learnable tokens, are encoded into positive/negative embeddings that guide the model to learn generalized facial authenticity patterns. In parallel, visual and audio encoders from a pre-trained lip-reading model provide audio-visual alignment features. A weight generator and modality modulator then fuse the alignment features with the learned facial patterns to produce the final detection score.

In T-AVFD, visual encoder $E_v$ and audio encoder $E_a$ are derived from the visual and auditory front-ends of the pre-trained lip-reading model. We first crop mouth regions from video frames $\{M_t\}_{t=0}^{T}$ and concatenate them along the temporal axis to form a structured visual sequence, which is subsequently encoded by $E_v$ into visual features. In parallel, Mel-spectrograms are computed from the audio $\{A_t\}_{t=0}^{T}$ and fed into $E_a$ to derive audio features. To enable cross-modal fusion with authentic facial patterns, visual and audio features are projected via linear layers into dimensionally aligned representations $v$ and $a$.

**Differential Weight Learning.** Most existing multi-modal forgery detectors adopt uniform fusion (Smeu et al., 2025; Feng et al., 2023), overlooking the modality-specific reliability variations under different forgery types. To address this, we introduce a differential weight learning mechanism that adjusts the contributions of audio $a$, visual $v$, and authentic facial patterns $fp$ via a weight generator and a modality modulator. We first use the weight generator to estimate the relative contributions of each modality:

$$\acute{w} = \delta\left(MLP\left(CAT\left[a, v, fp\right]\right)\right), \qquad (4)$$

where $\delta$ refers to the softmax function, $\acute{w} = \{\acute{w}_{fp}, \acute{w}_v, \acute{w}_a\}$ denotes the modality-dependent weights. $MLP$ and $CAT$ denote the multilayer perceptron and concatenation opera-

tion, respectively.

Although incorporating real face distributions can enhance generalization, audio-visual consistency remains an important signal for detecting forgeries in both talking and singing videos. To further guide modality fusion, we introduce a manually controlled modulation vector $\alpha = \{\alpha_1, \alpha_2, \alpha_3\}$ that emphasizes audio-visual alignment while preserving the ability to encode authentic facial patterns. In our implementation, we set $\alpha = \{-0.1, +0.1, +0.1\}$, corresponding to $\{authentic\ facial\ patterns, visual, audio\}$. This modulation modifies the fusion weights as:

$$w = \delta\left(\acute{w} + \alpha\right). \qquad (5)$$

The modulated weights $w = \{w_{fp}, w_v, w_a\}$ are multiplied with corresponding modality features to predict the final detection score $s$. Differential weight learning enables elegant feature fusion and strong generalization to the unseen singing domain, even when trained solely on real talking videos.

### 4.3. Training Objective

Following (Feng et al., 2023), we enhance temporal alignment between audio and visual frames by maximizing their correspondence probability. The audio-visual loss is defined

*Table 2.* **Results on the AVLips, FKAV, THB, and our SHDF.** We report AP (%) and AUC (%) with the best results in **bold** and the second-best results are underlined. *V* and *A* denote visual and audio modalities, respectively. *sup.* and *unsup.* refer to supervised and unsupervised methods. *talking* and *singing* are dataset types. We use released checkpoints for supervised baselines. Since AVAD has no training code, we evaluate it using its provided weights. AVH-Align is retrained under the same talking data for fair comparison.

| Methods | Type | Modality | AVLips (*talking*) | | FKAV (*talking*) | | THB (*talking*) | | SHDF (*singing*) | |
|---|---|---|---|---|---|---|---|---|---|---|
| | | | AP | AUC | AP | AUC | AP | AUC | AP | AUC |
| CViT (Wodajo & Atnafu, 2021) | sup. | V | 63.5 | 63.1 | 91.1 | 88.5 | 44.5 | 42.1 | 62.7 | 49.5 |
| EfficientViT (Coccomini et al., 2022) | sup. | V | 63.3 | 64.8 | 95.1 | 90.9 | 31.6 | 21.7 | 66.6 | 46.5 |
| RealForensics (Haliassos et al., 2022) | sup. | AV | 69.9 | 71.9 | 94.2 | 88.2 | 68.7 | 74.3 | 67.7 | 50.9 |
| LipFD (Liu et al., 2024b) | sup. | AV | **85.3** | 84.7 | 83.4 | 77.0 | 45.0 | 49.2 | 38.1 | 50.5 |
| AVAD (Feng et al., 2023) | unsup. | AV | 76.5 | 73.2 | 92.1 | 84.8 | 43.8 | 48.1 | 62.4 | 48.3 |
| AVH-Align (Smeu et al., 2025) | unsup. | AV | 74.3 | 84.5 | 93.5 | 93.0 | 64.8 | 82.3 | 55.2 | 37.4 |
| **T-AVFD (Ours)** | unsup. | AV | 83.6 | **87.7** | **95.6** | **95.6** | **87.6** | **93.0** | **85.7** | **80.2** |

as the negative average alignment probability across the entire video:

$$\mathcal{L}_{av} = -\frac{1}{F} \sum_{i=1}^{F} log \frac{e^{\Phi_{ii}}}{\sum_{k \in T_{(i)}} e^{\Phi_{ik}}}, \quad (6)$$

where $\Phi$ contains the audio frame $a_i$ and its corresponding video frame $v_i$, $T_{(i)}$ denotes the temporal neighborhood of the $i^{th}$ frame, $F$ is the frame number, and $k$ refers to a non-aligned frame. The final loss function is then defined as:

$$\mathcal{L} = \mathcal{L}_{av} + \mathcal{L}_{ft}. \quad (7)$$

We estimate the detection score for each audio-visual frame pair, where lower scores indicate stronger cross-modal coherence. To obtain a video-level assessment, we use a smoothed max operator to aggregate frame-wise scores. Let $s_t$ denote the frame-level score of the $t$-th frame. The video-level score is computed as:

$$s = \log \sum_{t=1}^{F} \exp(s_t). \quad (8)$$

## 5. Experiments

### 5.1. Experimental Setup

**Datasets.** Our experiments cover two distinct audio-visual scenarios, talking and singing, to evaluate the generalization capability of the proposed method. For the singing scenario, we conduct experiments on the proposed SHDF dataset. We adopt AVLips (Liu et al., 2024b), FakeAVCeleb (Khalid et al., 2021), and TalkingHeadBench (Xiong et al., 2025) for the talking scenario, encompassing a range of talking head generation models. Evaluation on the SHDF dataset is conducted using a test set comprising 800 real samples and 1,500 synthesized samples selected from the full dataset. It includes variations in gender, synthesis methods, and

speaker identities. In the evaluation on talking head datasets, the AVLips dataset is divided into training, validation, and test sets with a ratio of 6:1:3. For the FakeAVCeleb (FKAV) dataset, we construct a test set comprising 500 real samples and 1,000 carefully selected forged samples. The target test set for TalkingHeadBench (THB) is formed by aggregating all official test videos. To ensure a fair comparison of performance across both talking and singing scenarios, the proposed T-AVFD model is trained exclusively on real samples from the talking data, maintaining consistency in training data distribution. To assess whether detection performance depends on scenario-specific data distributions, we retrain AVH-Align (Smeu et al., 2025) and the proposed T-AVFD on the SHDF training set and evaluate them on both talking and singing data.

**Implementation Details.** The comparative methods include both supervised and unsupervised forgery detection models. We include CViT (Wodajo & Atnafu, 2021), EfficientViT (Coccomini et al., 2022), RealForensics (Haliassos et al., 2022), and LipFD (Liu et al., 2024b) as supervised baselines, and adopt AVAD (Feng et al., 2023) and AVH-Align (Smeu et al., 2025) as representative unsupervised models. Our model is trained using the Adam optimizer on a single NVIDIA A100 GPU with a learning rate of $9 \times 10^{-4}$. The batch size is set to 512, with the coefficients of $\mathcal{L}_{av}$ and $\mathcal{L}_{ft}$ both set to 1. Performance is evaluated using Average Precision (AP) and Area Under the Curve (AUC).

### 5.2. Experimental Results

**Cross-Dataset Generalization.** To evaluate the generalization capability of the proposed method, we conduct experiments on the talking datasets AVLips, THB, and FKAV, as well as the singing dataset SHDF. As shown in Table 2, all unsupervised methods are trained on real talking data, whereas supervised detectors rely on paired annotations of real-fake samples. In the talking scenario, T-AVFD achieves the best performance among nearly all detectors. This result

*Table 3.* **Comparison of models trained on singing data.** Results are reported on both *talking* (AVLips) and *singing* (SHDF) datasets using AP (%) and AUC (%). Our method (T-AVFD) demonstrates strong performance across both domains.

| Methods | AVLips | | SHDF | |
|---|---|---|---|---|
| | AP | AUC | AP | AUC |
| AVH-Align (Smeu et al., 2025) | 57.7 | 52.6 | 72.6 | 63.5 |
| **T-AVFD (Ours)** | **80.3** | **77.3** | **80.7** | **73.5** |

*Table 4.* **Comparison of textual-visual similarity across real and fake faces.** "P-Texts" and "N-Texts" correspond to positive and negative prompts. "Difference" is computed as the similarity gap. *real face* and *fake face* denote the types of extracted facial samples. A positive gap for real faces and a negative gap for fake faces indicate that Alpha-CLIP exhibits stronger sensitivity to real-fake texts than the original CLIP.

| Extractor | P-Texts | N-Texts | Difference |
|---|---|---|---|
| CLIP (*real face*) | 0.1865 | 0.1758 | +0.0107 |
| CLIP (*fake face*) | 0.1879 | 0.1860 | +0.0019 |
| Alpha-CLIP (*real face*) | 0.2138 | 0.1726 | **+0.0412** |
| Alpha-CLIP (*fake face*) | 0.1842 | 0.2067 | **-0.0225** |

validates the effectiveness of building an anomaly detection mechanism based on authentic facial patterns. It enables the model to learn more stable and generalizable features, thereby maintaining strong detection performance on data synthesized by various generative models, including GAN-based datasets (FKAV and AVLips) and diffusion-based ones (THB). In the singing scenario, all baseline methods yield AUC scores around 50%, while our method performs significantly better. This suggests that alignment-centric evidence learned in talking settings may be insufficient when the audio-visual regime shifts from talking to singing. In contrast, T-AVFD captures stable cross-scenario forgery detection features, demonstrating stronger generalization.

**Dependency on Training Data Distribution.** To evaluate dependence on the training data distribution, we retrain T-AVFD and AVH-Align (Smeu et al., 2025) on the singing dataset and test them on AVLips and SHDF. The training set consists of 1,500 real samples from SHDF that are disjoint from the test set.

As shown in Table 3, when trained on singing data, AVH-Align exhibits a drastic performance drop on AVLips, with AP and AUC scores approaching 50%, indicating near-undetectable results. In contrast, its performance on SHDF improves significantly, yielding distinguishable outcomes. These results suggest that AVH-Align is constrained by the training data distribution and fails to generalize across audio-visual scenarios with varying characteristics. In comparison, our method achieves competitive performance across different test scenarios regardless of whether it is trained on talking or singing data, demonstrating strong generalization.

**Effectiveness of Facial Authenticity Pattern.** To validate the effectiveness of our text-guided authentic facial pattern learning strategy, we input predefined positive and negative texts into the CLIP text encoder to obtain the corresponding textual features. The positive texts describe semantics related to authentic faces, while the negative texts contain descriptions associated with forged or abnormal facial attributes. We then use Alpha-CLIP to extract visual features from facial frames and compute their similarity to both positive and negative textual features. To further assess the necessity of Alpha-CLIP, we conduct a comparative experiment using the original CLIP model to extract facial features. Similarity is computed frame-by-frame against the textual features, and the average similarity is used as the final metric.

Experimental results in Table 4 show that Alpha-CLIP features exhibit more pronounced response differences between positive and negative texts, while the original CLIP produces smaller differences and consistently favors positive texts. These findings suggest that Alpha-CLIP, guided by positive-negative textual supervision, is sensitive to semantic distinctions between authentic and forged faces, thereby offering a discriminative feature foundation for audio-visual forgery detection.

**Robustness under Perturbations.** Manipulated audio-visual content is frequently subject to degradation resulting from platform-level processing and transmission. These perturbations may substantially compromise the reliability of forgery detection. Accordingly, evaluating model robustness under such conditions is essential for determining its practical applicability beyond idealized test environments. We introduce six types of perturbations: *Gaussian blur, JPEG compression, color inversion, Gaussian noise, pixelation, and image resizing*. To ensure comprehensive evaluation, these perturbations are applied to both the talking dataset THB (Xiong et al., 2025) and our singing dataset SHDF to evaluate model robustness under varied conditions.

As shown in Figure 5, our method consistently outperforms the baselines across talking and singing datasets, demonstrating strong resistance to perturbations. On the THB dataset, it achieves an average AUC of 84.6%, significantly surpassing AVAD (37.8%) and AVH-Align (43.2%). Notably, under blur, compression, and resize, our method maintains near-clean performance (AUC > 90%). Even under more challenging perturbations such as color inversion, pixelation, and noise, it still delivers reliable results. Similarly, on the SHDF dataset, our method exhibits robust performance, achieving an average AUC of 75.0%. These results confirm the strong robustness of our T-AVFD across diverse audio-visual scenarios and perturbation conditions.

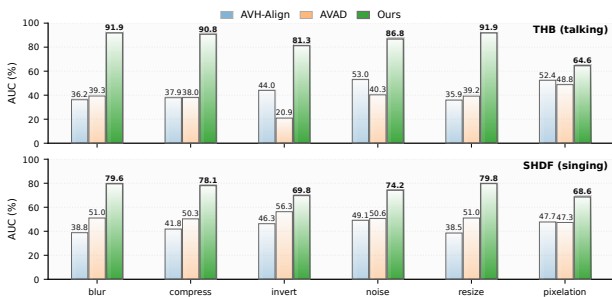

*Figure 5.* **Robustness evaluation under various perturbations.** We apply six types of perturbations to the singing and talking datasets for comparative evaluation.

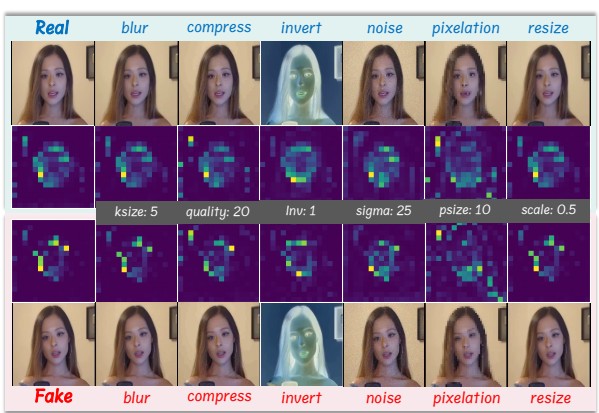

*Figure 6.* **Visualization of visual perturbations and their corresponding feature maps.** We apply six types of perturbations. *ksize, quality, inv, sigma, psize*, and *scale* denote the kernel size, JPEG quality, inversion flag, noise intensity, pixel block size, and resize ratio, respectively. Please zoom in for a detailed view.

To uncover the underlying mechanism behind the strong robustness of our method, we conduct a visualization in Figure 6. It reveals that, under various types of perturbations, semantic features of real faces remain significantly richer than those of forged ones. This supports the effectiveness of incorporating authentic facial pattern learning into the T-AVFD framework. Furthermore, we observe that under blur, compress, and resize, facial features are most consistent with the original reference, which explains the superior performance under these three perturbation types compared to the others.

**Singing-Induced Challenge Beyond the Generator.** To determine whether the detection challenge in singing is driven by the singing content regime itself or by generator-specific forgery signatures, we conduct a generator-controlled experiment. We fix the generator to MEMO and synthesize 500 talking and 500 singing videos for the same identities. As shown in Table 5, AVH-Align and AVAD suffer a clear performance drop on singing compared to talking. In contrast, T-AVFD is more stable, achieving 84.25/89.34 on talking

*Table 5.* **Generator-controlled talking vs. singing.** Results are reported with AP (%) and AUC (%).

| Methods | Talking | | Singing | |
|---|---|---|---|---|
| | **AP** | **AUC** | **AP** | **AUC** |
| AVAD (Feng et al., 2023) | 47.91 | 53.32 | 43.98 | 42.22 |
| AVH-Align (Smeu et al., 2025) | 76.17 | 88.00 | 44.06 | 42.40 |
| **T-AVFD (Ours)** | **84.25** | **89.34** | **76.74** | **79.95** |

*Table 6.* **Facial-region prompt analysis.** Results are reported with AP (%) and AUC (%).

| Prompt | SHDF (*singing*) | | THB (*talking*) | |
|---|---|---|---|---|
| | **AP** | **AUC** | **AP** | **AUC** |
| **face** | **80.5** | **73.0** | **80.2** | **91.1** |
| eyes | 77.6 | 68.9 | 78.1 | 89.6 |
| mouth | 74.1 | 67.2 | 74.5 | 88.9 |

and still 76.74/79.95 on singing (AP/AUC). These results indicate that the performance degradation in singing is primarily driven by the talking-to-singing domain shift, rather than the generator signature.

**Prompt Analysis.** Our prompt design follows a hierarchy built around face, eyes, and mouth. FAPL augments these prompts with learnable tokens, preserving predefined facial semantics while allowing the textual representation to adapt to the target feature distribution. To validate the role of each region in this hierarchy, we construct prompts based on face, eyes, and mouth separately. As shown in Table 6, face achieves the best overall results, followed by eyes and mouth. This comparison indicates that holistic facial semantics provide a more reliable textual prior than isolated articulatory-region descriptions, which are more sensitive to visual variations across speaking and singing.

We further examine the role of prompt learnability. As shown in Table 7, both removing learnable tokens and replacing all textual prompts with learnable tokens degrade performance. Fully fixed prompts limit adaptation to the target feature distribution, whereas fully learnable prompts discard the explicit facial-region structure. FAPL instead combines predefined multi-granular facial prompts with learnable tokens, retaining facial-region semantics while preserving adaptability.

### 5.3. Ablation Study

In Table 8, we perform a detailed ablation study on the SHDF (singing) and THB (talking) datasets to evaluate the contribution of each component in T-AVFD.

**Impact of Text Prompts.** Removing all text prompts (*w/o texts*) significantly reduces AP and AUC, while using a

*Table 7.* **Prompt learnability analysis.** Results are reported with AP (%) and AUC (%).

| Learnability | SHDF (*singing*) | | AVLips (*talking*) | |
|---|---|---|---|---|
| | AP | AUC | AP | AUC |
| No learnable tokens | 83.8 | 77.6 | 82.1 | 86.2 |
| All learnable tokens | 81.9 | 74.8 | 80.8 | 84.9 |
| **Ours** | **85.7** | **80.2** | **83.6** | **87.7** |

*Table 8.* **Ablation study of the proposed T-AVFD on SHDF and THB datasets.** We report AP (%) and AUC (%). Each ablation removes or modifies a key component to evaluate its contribution.

| Methods | SHDF (*singing*) | | THB (*talking*) | |
|---|---|---|---|---|
| | AP | AUC | AP | AUC |
| w/o texts | 74.6 | 62.0 | 75.2 | 89.5 |
| w/ single text | 80.5 | 73.0 | 80.2 | 91.1 |
| w/o face feature | 66.5 | 45.1 | 78.8 | 90.9 |
| w/o $\mathcal{L}_{ft}$ | 73.2 | 61.3 | 75.0 | 89.5 |
| w/o FAPL | 68.8 | 50.6 | 75.8 | 88.9 |
| w/o DWL | 76.3 | 68.7 | 66.0 | 80.4 |
| **T-AVFD (Full)** | **85.7** | **80.2** | **87.6** | **93.0** |

single text prompt (*w/ single text*) partially recovers performance. This demonstrates that multi-granular texts across face, eyes, and mouth are essential for guiding FAPL to learn fine-grained facial authenticity patterns.

**Analysis of Face Feature.** Ablating the face feature (*w/o face feature*) causes a dramatic drop in SHDF, indicating that the facial embedding is a critical source of forgery information. Interestingly, the THB performance remains relatively higher, suggesting that audio-visual cues alone can partially compensate in talking videos.

**Importance of Face-Text Contrastive Alignment.** Removing the FTCA loss (*w/o $\mathcal{L}_{ft}$*) results in similar degradation to *w/o texts*, which confirms that contrastive alignment between facial embeddings and positive-negative texts is necessary. $\mathcal{L}_{ft}$ enforces polarity-opposed supervision, enhancing the ability of FAPL to capture authentic patterns.

**Ablation of FAPL Module.** The complete removal of FAPL (*w/o FAPL*) further reduces performance. This validates that FAPL provides robust facial authenticity priors that complement audio-visual features, particularly enhancing cross-scenario generalization from talking to singing.

**Effect of Differential Weight Learning (DWL).** When DWL is removed (*w/o DWL*), the model shows a significant performance drop on both SHDF and THB datasets, indicating that uniform fusion strategies fail to accommodate modality-specific reliability differences across forgery types. In contrast, DWL dynamically adjusts modality weights, allowing FAPL patterns and audio-visual alignment to work

synergistically for forgery detection, especially critical in unseen singing videos, where audio-visual synchronization is less reliable.

## 6. Conclusion

In this paper, we have expanded the scenario of audio-visual deepfake detection from talking to singing, aiming to facilitate a more challenging assessment of model generalization and robustness. To this end, we construct SHDF, the first audio-visual deepfake dataset specifically designed for singing scenarios. We further identify consistent semantic discrepancies between real and forged facial content. It motivates the design of T-AVFD, a text-guided unsupervised detection framework that learns semantic patterns from authentic faces. Experiments on both talking and singing datasets show that our method achieves strong performance with robust generalization.

## Impact Statement

This paper presents work whose goal is to advance the field of machine learning. There are many potential societal consequences of our work, none of which we feel must be specifically highlighted here.

## Acknowledgments

This work was supported in part by the Fundamental and Interdisciplinary Disciplines Breakthrough Plan of the Ministry of Education of China under Grant JYB2025XDXM102, and in part by the National Natural Science Foundation of China under Grant 62220106008.

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

# A. Appendix

## A.1. Overview

The supplementary material contains **four** subsections, specifically as follows:

- **More Details of SHDF Dataset.** We outline the dataset synthesis methods, show representative samples, conduct a comparative analysis between the singing and talking datasets, and include the data availability statement.

- **Additional Experimental Details.** We summarize the comparison methods, introduce the talking head datasets, present the perturbation settings, explain the modulation vector configurations, and discuss the inference efficiency.

- **Extended Experiments.** We report results under different modulation vector configurations and diverse positive–negative text pairings.

- **Discussion and Future Challenges.** We point out current limitations and highlight open challenges, emphasizing directions for future research.

## A.2. More Details of SHDF Dataset

**Dataset Synthesis Methods.** Existing audio-visual deepfake datasets are predominantly designed for talking scenarios, whereas singing, despite being a significant and representative form of audio-visual media, has been largely overlooked. To fill this data gap, we construct a singing deepfake dataset using rhythm-aware audio-visual synthesis models, aiming to provide a more challenging benchmark for research on audio-visual forgery detection. Table 9 presents current rhythm-aware singing head synthesis methods. Among those with publicly available code, we select MEMO (Zheng et al., 2024), EchoMimic (Chen et al., 2025), Hallo2 (Cui et al., 2025a), and DreamTalk (Ma et al., 2023) as candidate synthesizers. Due to the expensive computational demands, Hallo3 (Cui et al., 2025b) and InfiniteTalk (Yang et al., 2025) are not selected for inclusion.

Given that audio-visual forgery content is typically designed to directly target end users, we conduct a comprehensive user study to better evaluate the quality of videos generated by different synthesizers. For each method, we randomly select 30 videos to ensure coverage of diverse identities, songs, and visual content. A total of 20 participants evaluate each video based on three criteria: lip sync accuracy, visual quality, and head motion. The evaluation uses a 5-point scale, where 1 indicates the worst and 5 indicates the best. The final score for each video is calculated by averaging the ratings from all participants. Subsequently, for each method, we compute the average score across its sampled videos on each criterion to obtain the final performance score. As shown in Table 10, DreamTalk performs poorly in terms of head motion, which is particularly detrimental for singing heads that require rich and expressive movements. Therefore, we ultimately select MEMO, EchoMimic, and Hallo2 as our synthesizers.

*Table 9.* Overview of singing head methods including publication venue, year, and code availability.

| Method | Venue | Code Availability |
|---|---|---|
| EMO (Tian et al., 2024) | ECCV'24 | No |
| EMO2 (Tian et al., 2025) | ArXiv'25 | No |
| Hallo2 (Cui et al., 2025a) | ICLR'25 | Yes |
| Hallo3 (Cui et al., 2025b) | CVPR'25 | Yes |
| MEMO (Zheng et al., 2024) | ArXiv'24 | Yes |
| EchoMimic (Chen et al., 2025) | AAAI'25 | Yes |
| DreamTalk (Ma et al., 2023) | ArXiv'23 | Yes |
| InfiniteTalk (Yang et al., 2025) | ArXiv'25 | Yes |

MEMO mitigates temporal error accumulation and enhances long-term identity consistency through a memory-guided temporal module. Leveraging an emotion-aware audio module, it achieves precise alignment between audio, lip movements, and facial expressions. Hallo2 is a latent diffusion-based method for audio-visual portrait video synthesis. It supports long-duration, high-resolution generation while maintaining temporal coherence and accurate lip synchronization. By

*Table 10.* Average user ratings (1–5) on Lip Sync (LS), Visual Quality (VQ), and Head Motion (HM) for the evaluated singing head synthesis methods. The best is indicated with **bold**, and the second-best is indicated with underline.

| Method | LS | VQ | HM | Selected |
|---|---|---|---|---|
| MEMO (Zheng et al., 2024) | 4.6 | **4.5** | **4.4** | ✓ |
| Hallo2 (Cui et al., 2025a) | **4.8** | 4.2 | 4.0 | ✓ |
| EchoMimic (Chen et al., 2025) | 4.3 | 4.1 | 4.1 | ✓ |
| DreamTalk (Ma et al., 2023) | 4.2 | 3.8 | 1.9 | × |

*Table 11.* Generator-side facial authenticity evaluation. Average user ratings are reported on a 1-5 scale.

| Generator | Facial Naturalness | Cross-Region Coherence | Temporal Consistency |
|---|---|---|---|
| EchoMimic (Chen et al., 2025) | 3.3 | 3.6 | 4.0 |
| Hallo2 (Cui et al., 2025a) | 3.5 | 3.3 | 3.7 |
| DreamTalk (Ma et al., 2023) | 2.3 | 3.8 | 3.9 |
| MEMO (Zheng et al., 2024) | **4.4** | **4.5** | **4.2** |

conditioning on audio input, Hallo2 generates head movements that preserve identity. EchoMimic is an audio-driven portrait animation method that integrates audio signals and facial landmarks to generate high-quality and expressive audio-visual talking head videos. Through an innovative training strategy, EchoMimic significantly improves the realism and visual appeal of generated animations. Equipped with key capabilities such as prosody awareness, precise lip synchronization, identity preservation, and responsive motion, MEMO, Hallo2, and EchoMimic are particularly well-suited for high-quality singing head synthesis.

To further assess the realism of generated facial content, we additionally introduce generator-side facial authenticity as an evaluation dimension. Specifically, we evaluate each generator from three aspects: facial naturalness, cross-region coherence, and temporal consistency. Facial naturalness measures frame-level facial realism, cross-region coherence evaluates whether different facial regions remain visually consistent, and temporal consistency reflects the stability of facial appearance across frames. As shown in Table 11, MEMO achieves the highest scores across all three dimensions, indicating stronger facial authenticity among the evaluated generators. Therefore, we use MEMO-based singing samples as a more challenging evaluation setting. As reported in Table 5, all methods degrade under this setting, suggesting that higher generator-side facial authenticity increases the difficulty of singing deepfake detection. Nevertheless, T-AVFD still achieves the best overall performance, indicating that it remains effective when synthesized samples exhibit more realistic facial appearances.

**Visualization of Synthesized Singing Heads.** Figure 7 presents representative synthesis examples from SHDF. Compared to conventional talking-head datasets, singing scenarios typically involve more intense facial expressions, more complex mouth movements, and greater motion dynamics. SHDF provides high-fidelity synthesized videos that capture these challenging audio-visual conditions, offering valuable benchmark data for comprehensive and reliable evaluation of audio-visual forgery detection methods.

**Comparative Analysis of Audio-Visual Datasets.** As shown in Table 12, most existing datasets focus exclusively on talking scenarios and differ in terms of manipulated modality, number of generators, dataset scale, and quality verification. In contrast, SHDF expands the talking head paradigm to singing scenarios, which feature richer facial expressions and more complex mouth movements than traditional talking videos. This singing deepfake dataset guarantees high-fidelity synthesis and includes comprehensive quality verification, comprising 2,600 real videos and 3,000 generated videos produced by multiple generators. Compared to existing talking head datasets, SHDF provides a novel and more challenging benchmark for evaluating audio-visual deepfake detection methods, contributing to the mitigation of harmful content dissemination.

**Data Availability Statement.** The proposed SHDF dataset comprises two components: real singing videos collected from publicly accessible online platforms and forged videos synthesized through our custom generation pipeline. For the real videos, we will provide only public YouTube links, while the original video files will not be redistributed. This design is intended to avoid unauthorized redistribution of third-party content and to respect the original platform terms and applicable copyright regulations. The dataset is released solely for academic research on deepfake detection. Users are responsible for accessing and using the referenced videos in accordance with YouTube's Terms of Service, applicable copyright laws, and relevant data protection regulations.

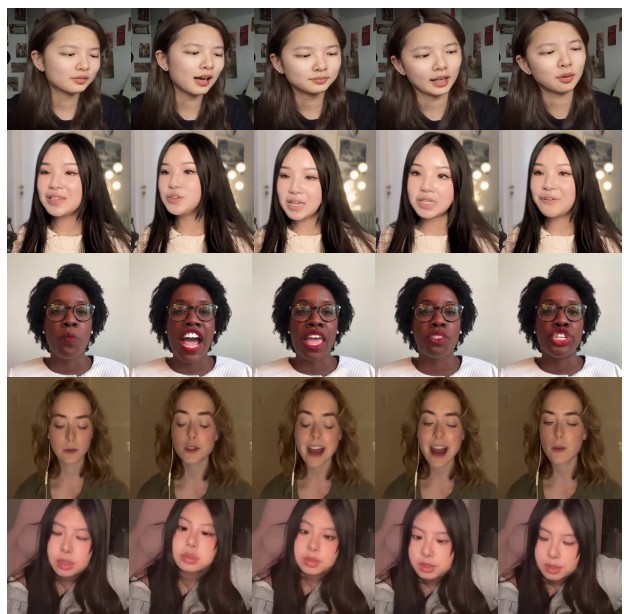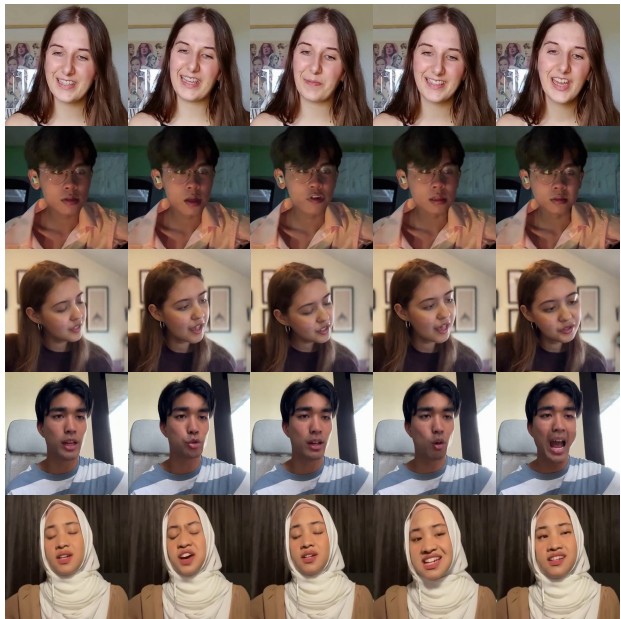

*Figure 7.* Sample frames from our SHDF dataset. The examples show individuals across different genders and identities.

*Table 12.* Comparison of audio-visual datasets for talking heads and singing heads. *Manipulated Modality* indicates the modality involved in manipulation, *Scenario* indicates the type of scene represented, and *Verified* denotes whether the data has undergone quality inspection. SHDF provides high-fidelity singing head samples that are absent in existing datasets, offering a more challenging benchmark for audio-visual deepfake detectors.

| Dataset | Venue | Manipulated Modality | Scenario | #Generators | #Real | #Fake | Verified |
|---|---|---|---|---|---|---|---|
| FaceForensics++ (Rossler et al., 2019) | ICCV'19 | Visual | Talking | Multiple | 1,000 | 4,000 | ✓ |
| FakeAVCeleb (Khalid et al., 2021) | NIPS'21 | Audio/Visual | Talking | Multiple | 500 | 19,500 | ✓ |
| LAV-DF (Cai et al., 2022) | DICTA'22 | Audio/Visual | Talking | Single | 36,431 | 99,873 | ✗ |
| AV-Deepfake1M (Cai et al., 2024) | ACM MM'24 | Audio/Visual | Talking | Single | 286,721 | 860,039 | ✗ |
| DF-40 (Yan et al., 2024) | NIPS'24 | Visual | Talking | Multiple | 1,590 | 1M+ | ✗ |
| AVLips (Liu et al., 2024b) | NIPS'24 | Audio/Visual | Talking | Multiple | 20,000+ | 340,000 | ✗ |
| TalkingHeadBench (Xiong et al., 2025) | ArXiv'25 | Audio/Visual | Talking | Multiple | 2,312 | 2,984 | ✓ |
| SHDF (Ours) | - | Audio/Visual | Singing | Multiple | 2,600 | 3,000 | ✓ |

### A.3. Additional Experimental Details

**Baseline Detectors.** We adopt both supervised and unsupervised forgery detectors as baselines in our paper. The unsupervised methods, including AVAD (Feng et al., 2023) and AVH-Align (Smeu et al., 2025), are trained solely on real talking head data. In contrast, the supervised models are trained on both real and manipulated talking head samples, comprising CViT (Wodajo & Atnafu, 2021), RealForensics (Haliassos et al., 2022), EfficientViT (Coccomini et al., 2022), and LipFD (Liu et al., 2024b).

AVAD is an audio-visual deepfake detection method that assesses temporal inconsistencies between speech and facial motion. By analyzing whether the observed desynchronization conforms to natural patterns or exhibits abnormal deviations, the method determines the authenticity of a video. AVH-Align identifies a common artifact in forged videos (leading silent segments) that can inadvertently bias existing detection models. To mitigate this issue, it proposes an unsupervised alignment strategy that discourages models from relying on such dataset-specific cues, thereby improving their robustness against a broader range of manipulations.

Regarding supervised detection models, CViT is a vision-based deepfake detector that combines convolutional neural networks with Transformer-based attention mechanisms, enabling the model to capture both fine-grained local features and global structural patterns in manipulated frames. RealForensics leverages the natural correspondence between visual and auditory modalities in real videos. By employing a self-supervised cross-modal learning framework, it learns temporally

dense video representations that encode facial dynamics, expressions, and identity-related cues. EfficientViT adopts a hybrid architecture that integrates multiple variants of vision transformers with a lightweight EfficientNet-B0 backbone for feature extraction, achieving strong performance in video forgery detection while maintaining computational efficiency. LipFD is a supervised audio-visual detector specifically designed for talking-head scenarios that identifies manipulated content by modeling inconsistencies between the speech signal and the corresponding lip movements.

**Talking Head Benchmarks.** In addition to our proposed SHDF singing head dataset, we conduct experiments on three representative talking head datasets: FakeAVCeleb (FKAV) (Khalid et al., 2021), AVLips (Liu et al., 2024b), and TalkingHeadBench (THB) (Xiong et al., 2025). These datasets reflect typical distributions and synthesis paradigms found in talking head data. This comprehensive evaluation enables us to assess the generalization capabilities of detection models across both talking and singing scenarios.

The FKAV dataset consists of 500 real videos sourced from VoxCeleb2 (Chung et al., 2018) and 19,500 forged videos generated using a variety of manipulation techniques. These include visual forgeries via Faceswap (Korshunova et al., 2017), FSGAN (Nirkin et al., 2019), and Wav2Lip (Prajwal et al., 2020), as well as audio forgeries synthesized using SV2TTS (Jia et al., 2018). We address the imbalance between real and fake samples by constructing a test set comprising 500 real videos (excluded from all training sets) and 1,000 fake videos generated using diverse forgery methods. AVLips is a large-scale audio-visual lip-sync dataset specifically designed for talking head forgery detection. It contains approximately 340,000 audio-video samples synthesized using various lip-syncing approaches. To simulate realistic lip movements, the dataset incorporates both static generation methods such as MakeItTalk (Zhou et al., 2020) and dynamic techniques including Wav2Lip, TalkLip (Wang et al., 2023), and SadTalker (Zhang et al., 2023). Real samples are primarily drawn from LRS3 (Afouras et al., 2018), FaceForensics++ (Rossler et al., 2019), DFDC (Dolhansky et al., 2020), and other real-world sources. The full dataset is not publicly available. Instead, the authors provide a subset that contains real samples from LRS3 and synthetic videos generated using unspecified GAN-based methods. We partition this subset into training, validation, and test sets with a 6:1:3 ratio. Both our method and the unsupervised AVH-Align baseline are trained on real samples from the training split. THB is a recently introduced dataset for audio-visual forgery detection, primarily constructed using diffusion-based generation methods. It synthesizes manipulated videos by combining portrait images from FFHQ (Karras et al., 2019) with driving signals from CelebV-HQ (Zhu et al., 2022). The dataset includes samples produced by six representative methods: Hallo (Xu et al., 2024), Hallo2 (Cui et al., 2025a), AniPortrait (Audio-driven and Video-driven) (Wei et al., 2024), LivePortrait (Guo et al., 2024), and EMOPortraits (Drobyshev et al., 2024). Additionally, THB incorporates commercial samples generated by MAGI-1 (Sand-AI, 2025), a diffusion-based tool. The official repository provides separate training, validation, and test sets for each generation method. For evaluation, we aggregate all test samples generated by diffusion-based methods into a unified test set. Specifically, this set includes 117 videos from Hallo, 105 from Hallo2, 146 from AniPortrait (Audio-driven), 108 from AniPortrait (Video-driven), and 66 commercial samples from MAGI-1.

*Table 13.* Corruption settings applied to video frames for robustness evaluation. Each perturbation simulates real-world degradations.

| Corruption Type | Parameter | Range | Value |
|---|---|---|---|
| JPEG Compression (`compress`) | quality | 0–100 | 20 |
| Resize (`resize`) | scale | 0–1 | 0.5 |
| Gaussian Blur (`blur`) | ksize | odd integer | 5 |
| Gaussian Noise (`noise`) | $\sigma$ | $\geq 0$ | 25 |
| Color Inversion (`invert`) | inv | N/A | N/A |
| Pixelation (`pixelation`) | block size | {2, 4, 8, 10, 16} | 10 |

**Perturbation Settings.** In-the-wild videos are often vulnerable to various corruptions, which pose challenges for reliable deepfake detection. A robust detector must not only generalize well across datasets but also withstand typical perturbations to accurately identify manipulated content. To evaluate this, we test model performance under six representative perturbation types:

1. **JPEG Compression**: Simulates video encoding degradation by introducing detail loss and compression artifacts. Higher compression levels result in lower visual quality and more pronounced distortions.

2. **Resizing**: Reduces spatial resolution and blurs fine-grained structures. Smaller scales lead to increased blurring and loss of subtle facial details.

*Table 14.* Efficiency comparison. Training and inference costs are measured on 3,000 samples. "–" indicates that the training code is not publicly available, so the corresponding cost cannot be reliably estimated.

| Method | Train Memory | Train Time | Inference Memory | Inference Time |
|---|---|---|---|---|
| AVAD (Feng et al., 2023) | – | – | ~3.1GB | ~80min |
| AVH-Align (Smeu et al., 2025) | ~3GB | ~28min | ~1.3GB | ~1min |
| T-AVFD | ~4GB | ~36min | ~1.7GB | ~1.2min |

3. **Gaussian Blur**: Smooths high-frequency components, potentially masking fine lip movements and facial expressions. Larger kernel sizes produce stronger blurring effects.

4. **Gaussian Noise**: Mimics sensor or transmission errors by adding random pixel-level variations. Increasing the standard deviation ($\sigma$) intensifies the noise level.

5. **Color Inversion**: Alters the visual appearance by reversing pixel values while preserving spatial structure, testing the model's ability to handle extreme color shifts.

6. **Pixelation**: Aggregates neighboring pixels into coarse blocks, reducing local spatial resolution and obscuring fine details such as lip contours and expression dynamics. Larger block sizes result in stronger pixelation.

The configurations for each perturbation are summarized in Table 13.

**Modulation Vector Configuration.** To effectively detect forgeries in both talking head and singing head videos, we introduce a manually designed modulation vector $\alpha = \{-0.1,\, 0.1,\, 0.1\}$, which adjusts the contributions of authentic facial patterns ($fp$), visual features ($v$), and audio features ($a$) during multi-modal fusion. Let the original fusion weights be $\acute{w} = \{\acute{w}_{fp}, \acute{w}_v, \acute{w}_a\}$. After applying the modulation, the adjusted weights are:

$$\acute{w}' = \acute{w} + \alpha = \{\acute{w}_{fp} - 0.1,\; \acute{w}_v + 0.1,\; \acute{w}_a + 0.1\}. \tag{9}$$

The final modality-dependent fusion weights are obtained via:

$$w_i = \frac{\exp(\acute{w}_i')}{\sum_{j \in \{fp,v,a\}} \exp(\acute{w}_j')}, \quad i \in \{fp, v, a\}. \tag{10}$$

This modulation setting balances the contributions of static identity cues and dynamic audio-visual synchronization, which is crucial for robust detection across both talking head and singing head scenarios. The negative offset for $fp$ prevents the model from over-relying on facial appearance, which may vary significantly in expressive singing videos, while the positive offsets for $v$ and $a$ enhance sensitivity to audio-visual alignment, the most reliable cue for detecting forgeries in both modalities. By applying this differential modulation before the softmax normalization, the model effectively emphasizes the most informative modalities without disregarding identity-related information, leading to superior generalization performance across different audio-visual forgery scenarios. In the following section, we present an analysis of how different modulation vector configurations affect model performance.

**Inference Efficiency.** Feature extraction with Alpha-CLIP and the lip-reading network is performed offline. On an NVIDIA A100 GPU, processing 3,000 samples takes 35 minutes and 5 GB memory. With cached features, inference on the same 3,000 samples runs in 1.2 minutes with 1.7 GB memory. Training (excluding feature extraction) uses 4 GB memory and converges within 0.6 hours. These costs indicate that T-AVFD is practical for real-world use, with efficient batch inference once features are cached.

We compare the computational cost of T-AVFD with AVAD and AVH-Align, which are also trained using real data only. Following the same evaluation protocol, we report the training and inference cost on 3,000 samples. As shown in Table 14, T-AVFD requires slightly higher training and inference memory than AVH-Align, but remains within a comparable computational range. In terms of runtime, our method also maintains competitive efficiency. Combined with the detection results in Table 2, this comparison shows that T-AVFD achieves stronger detection performance without introducing excessive computational overhead.

*Table 15.* Experiments on different modulation vectors. Each modulation vector corresponds to weights applied to *[visual, audio, authentic facial patterns]* features. Results are reported on both *talking* (AVLips) and *singing* (SHDF) datasets using AP (%) and AUC (%), highlighting the impact of feature modulation on model performance across different scenarios. The best is indicated with **bold**, and the second-best is indicated with underline.

| Modulation Vector | AVLips (*talking*) | | SHDF (*singing*) | |
|---|---|---|---|---|
| | AP | AUC | AP | AUC |
| $[-0.1, -0.1, +0.1]$ | 77.5 | 83.4 | 69.9 | 59.6 |
| $[+0.1, +0.1, +0.1]$ | **87.9** | 89.4 | 81.5 | 74.5 |
| $[+0.1, -0.1, -0.1]$ | 80.4 | 86.2 | 83.7 | 78.7 |
| $[+0.1, -0.1, +0.1]$ | 87.4 | 87.8 | 72.8 | 63.5 |
| $[-0.1, +0.1, -0.1]$ | 87.8 | **90.1** | 79.1 | 71.6 |
| $[-0.1, +0.1, +0.1]$ | 79.8 | 85.6 | 84.3 | 79.3 |
| $[-0.1, -0.1, -0.1]$ | 79.7 | 85.5 | 75.4 | 70.2 |
| $[+0.1, +0.1, -0.1]$ | 83.6 | 87.7 | **85.7** | **80.2** |

*Table 16.* Analysis of the modulation vector $\alpha$. Results are reported with AP (%) and AUC (%).

| Modulation Vector | AVLips (*talking*) | | SHDF (*singing*) | |
|---|---|---|---|---|
| | AP | AUC | AP | AUC |
| $[0, 0, 0]$ | 81.4 | 85.8 | 82.2 | 76.3 |
| Learnable | 83.0 | 86.2 | 84.1 | 78.0 |
| Ours | **83.6** | **87.7** | **85.7** | **80.2** |

## A.4. Extended Experiments

**Different Modulation Vectors.** Table 15 presents the performance of different modulation vector settings on the talking (AVLips) and singing (SHDF) datasets. The model shows greater sensitivity to visual and audio feature modulation in talking scenarios, while in singing scenarios, the regulation of the authentic facial pattern has a more pronounced impact on generalization.

We find $\alpha = [+0.1, +0.1, -0.1]$ achieves consistently strong performance across both scenarios, striking a balance between talking and singing detection. Enhancing visual and audio features facilitates the capture of audio-visual synchronization cues, while moderately suppressing the authentic facial pattern helps mitigate overfitting to the training distribution, thereby improving generalization in the singing domain. In contrast, $\alpha = [+0.1, +0.1, +0.1]$ yields good results in the talking scenario but suffers a noticeable drop in AUC on the singing dataset, suggesting that excessive reliance on the authentic facial pattern limits adaptability to more complex facial dynamics. The configuration $\alpha = [-0.1, -0.1, +0.1]$ performs poorly in the singing scenario and shows the weakest performance in the talking domain. This suggests that although enhancing the authentic facial pattern may improve generalization, suppressing visual and audio features significantly undermines overall detection capability, especially in identifying audio-visual synchronization inconsistencies. Other combinations, such as $\alpha = [-0.1, +0.1, +0.1]$ and $\alpha = [+0.1, -0.1, +0.1]$, show moderate performance in one scenario but lack overall balance.

We further analyze the role of the modulation prior $\alpha$. As shown in Table 16, setting $\alpha$ to zero degrades performance, indicating that removing the modulation prior weakens cross-scenario adaptation. Learning $\alpha$ automatically improves over this setting, but still underperforms our adopted configuration. These results suggest that the proposed $\alpha$ provides an empirically effective prior for balancing authentic facial patterns, visual information, and audio information across different audio-visual scenarios.

**Different Positive-Negative Text Pairs.** To examine the impact of positive and negative text pairs on performance, we conduct comparative experiments with different pairing strategies, as shown in Table 17. The results demonstrate that the ratio of positive to negative pairs directly influences the learning of authentic facial patterns and the generalization ability in both talking and singing scenarios. When positive prompts dominate, the model performs better on the singing dataset (SHDF), suggesting that emphasizing positive supervision helps capture genuine facial features, which is beneficial for handling the rich expressions and mouth movements in singing. However, the performance on the talking dataset (AVLips) declines, which may be due to insufficient negative supervision. This causes the model to overemphasize facial semantics, weakening its ability to detect subtle audio-visual desynchronization. Conversely, when negative prompts dominate, the

model achieves higher performance on the talking dataset but degrades on the singing dataset. This indicates that negative supervision helps suppress false responses to forged features, though excessive emphasis may hinder the learning of authentic facial patterns.

*Table 17.* Comparative experiments with different text pairings. *pos.* and *neg.* refer to positive and negative text prompts, respectively, with the preceding number indicating the quantity of prompts. The best results are shown in **bold**, and the second-best results are underlined.

| Text Setting | AVLips (*talking*) | | SHDF (*singing*) | |
|---|---|---|---|---|
| | **AP** | **AUC** | **AP** | **AUC** |
| $(1pos., 3neg.)$ | 80.5 | 86.6 | 79.6 | 73.4 |
| $(3pos., 1neg.)$ | 76.1 | 85.1 | 82.2 | 77.1 |
| $(2pos., 2neg.)$ | 75.6 | 83.9 | 82.0 | 75.8 |
| $(4pos., 4neg.)$ | 76.0 | 85.0 | 80.9 | 74.4 |
| Ours $(3pos., 3neg.)$ | **83.6** | **87.7** | **85.7** | **80.2** |

For balanced text pair configurations, both insufficient and excessive prompt quantities tend to induce scenario-specific biases, thereby compromising overall generalization. In contrast, the configuration adopted in our T-AVFD yields optimal performance across both scenarios. This suggests that a moderately balanced ratio of positive and negative prompts facilitates the learning of authentic facial patterns, while also maintaining sensitivity to forged patterns and enabling robust generalization across diverse conditions.

We further evaluate whether our method is sensitive to the source of the predefined prompts. Under the same multi-granular prompt structure, we compare manually written prompts with prompts generated by Gemini (Team et al., 2023) and ChatGPT (Achiam et al., 2023). As shown in Table 18, ChatGPT-generated prompts achieve the best overall performance, while Gemini-generated and manual prompts also remain effective. This comparison indicates that our method does not rely on a single specific wording, although the quality and specificity of the predefined textual descriptions can still influence the final performance.

*Table 18.* Prompt source analysis. Results are reported with AP (%) and AUC (%) under the same multi-granular prompt structure.

| Prompt Source | THB (*talking*) | | SHDF (*singing*) | |
|---|---|---|---|---|
| | **AP** | **AUC** | **AP** | **AUC** |
| Manual | 83.3 | 88.9 | 81.9 | 74.1 |
| Gemini | 85.4 | 91.2 | 82.6 | 77.0 |
| ChatGPT | **87.6** | **93.0** | **85.7** | **80.2** |

## A.5. Discussion and Future Challenges

Most existing audio-visual forgery detection datasets primarily focus on talking scenarios, with little to no coverage of singing contexts. This limitation undermines the comprehensiveness and reliability of current detection systems in diverse real-world applications. To address this gap, we introduce the SHDF singing dataset, which offers a novel audio-visual setting and facilitates the advancement of forgery detection under more complex and expressive conditions. In terms of forgery detectors, unlike existing approaches that rely heavily on audio-lip synchronization features, our method learns generalized authentic facial patterns, enabling robust cross-scenario forgery detection with improved generalization.

However, current audio-visual forgery detection methods are typically limited to binary classification (i.e., real or fake) and lack the capability to trace the origin of the forgery, such as identifying the specific generation algorithm or manipulation technique used. This limitation reduces their utility in forensic applications, where source attribution is critical for accountability, legal investigation, and the development of targeted defense strategies.

We plan to extend our research in the following two directions:

**In forgery attribution**, we aim to explore multi-modal recognition methods based on generative features. By analyzing fine-grained artifacts in audio-visual data, model-specific statistical anomalies, and localized motion patterns, we seek to distinguish between different generation algorithms or manipulation techniques. This research will contribute to building a traceable forgery attribution framework, enhancing the precision and reliability of audio-visual forensic analysis.

**In dataset construction**, we plan to expand the SHDF by generating higher-quality deepfake data. This includes more diverse facial expressions and mouth movements, while also incorporating a broader range of generation models and synthesis strategies. Such high-fidelity data will provide a more challenging training and evaluation environment for forgery detectors, promoting robust generalization in complex scenarios.

