# OpenReview forum: "From Talking to Singing: A New Challenge for Audio-Visual Deepfake Detection"
_ICML.cc/2026/Conference — ICML 2026 regular_

### Official Review · Reviewer_bUiz · 2026-03-02

**Soundness:** 3
**Presentation:** 3
**Significance:** 2
**Originality:** 2
**Overall Recommendation:** 4
**Confidence:** 4

**Summary:**

This paper studies a practical but underexplored setting for audio-visual deepfake detection: transferring detectors trained on talking-head videos to singing videos. The authors show that singing introduces a substantially larger domain shift, where common lip–speech synchronization cues become less reliable and real/fake score distributions overlap more. To address this, they introduce a new singing-focused benchmark (SHDF) and propose T-AVFD, which complements an audio-visual alignment signal with a face-semantic authenticity pattern learned via vision–language prompting, and then fuses the two signals with a differential weighting scheme for final detection. Experiments across multiple talking datasets and the proposed singing benchmark suggest improved cross-domain robustness compared to prior baselines, including under common visual corruptions.

**Compliance With Llm Reviewing Policy:**

Affirmed.

**Final Justification:**

My main concerns are adequately addressed, so I keep the originally positive score.

**Key Questions For Authors:**

Please refer to the weaknesses section.

**Limitations:**

The paper’s discussion of limitations and societal risks is quite high-level for a deepfake detection paper that introduces a new singing deepfake dataset, and it does not concretely address dual-use/misuse concerns, data sourcing/consent, or deployment risks.

**Strengths And Weaknesses:**

**Strengths**

1. The paper targets a realistic failure mode of existing AV deepfake detectors and supports the motivation with quantitative evidence of reduced separability in singing videos.

2. The SHDF dataset provides a dedicated evaluation bed for singing deepfake detection, which can help the community study cross-domain generalization beyond the standard talking-head setting.

**Weaknesses**

1. The method relies on ChatGPT-generated positive/negative prompts (with face/eyes/mouth variants), yet the paper does not provide the complete prompt list or a deterministic generation/selection procedure.

2. The paper introduces a “manually controlled” modulation vector $\( \alpha=\{\alpha_1,\alpha_2,\alpha_3\} \)$ but does not report the numeric values used in the main experiments or the selection protocol.

3. The “smoothed max” video aggregation is undefined. The paper states that video-level predictions are obtained using a smoothed max operator over frame-wise scores, but it does not define the operator mathematically

4. One part of the dataset description claims EchoMimic and Hallo3 each contribute 500 forged samples, while another part states Hallo3 is not included due to high compute cost. This inconsistency undermines confidence in the dataset specification and affects claims about generator diversity.

5. The paper indicates several supervised baselines are evaluated from released checkpoints/provided weights, while the proposed approach benefits from strong pretrained components plus an added facial-semantic cue. When many baselines are near-random on SHDF, it is unclear whether the comparison reflects algorithmic improvement or simply lack of target-domain adaptation.

---

> ### Author Rebuttal · Authors · 2026-03-30
>
> **We sincerely thank Reviewer bUiz for the valuable comments.** We greatly appreciate and are encouraged by the reviewer’s recognition of our motivation, benchmark contribution, and the value of our problem setting. Below, we respond point by point to the concerns raised and will make the corresponding revisions in the revised manuscript.
>
> >**Q1:** The paper does not provide the complete prompt list or a deterministic generation procedure.
>
> **R:** Thank you for the comment. The prompt list is already described in Sec. 4.1 (Page 4), and we will further present it as a table in the revision for clearer reference.
>
> The prompt construction follows a structured procedure rather than unconstrained prompt search. Prior work [1] supports our prompt design based on the face, eyes, and mouth for deepfake detection. Based on this formulation, we use a fixed prompt template to construct candidate positive/negative prompt sets with progressively finer granularity, including 1 pair for face, 2 pairs for face+eyes, 3 pairs for face+eyes+mouth, and 4 pairs for face+eyes+mouth+skin. Table 5 reports the face-level single-pair setting, while Table 11 further evaluates the 2-, 3-, and 4-pair settings and shows that the 3-pair configuration gives the best overall balance across talking and singing. Therefore, the prompt set used in the main experiments is determined through a structured prompt construction process and empirical comparison. *[1] Towards More General Video-based Deepfake Detection through Facial Component Guided Adaptation for Foundation Model, CVPR 2025.*
>
> >**Q2:** The paper does not report the numeric values of &alpha; used in the main experiments or the selection protocol.
>
> **R:** Thank you for the comment. We have in fact specified in Appendix A.3.4 that the main experiments use &alpha;=[-0.1,+0.1,+0.1], corresponding to [authentic facial patterns, visual, audio]. We will state this explicitly in the main text.
>
> As analyzed in Appendix A.4.1, we evaluate multiple &alpha; configurations and compare their effects on both talking and singing. The results show that different settings lead to different balances between talking and singing, and [-0.1,+0.1,+0.1] gives the best overall trade-off.
>
> To further strengthen this point, we have compared &alpha; = 0, learnable &alpha;, and our &alpha;. &alpha;=0 degrades performance, while learnable &alpha; improves on this setting but remains below ours. The results indicate that the adopted &alpha; is not an arbitrary manual setting, but a manually specified and experimentally validated prior for cross-scenario transfer.
> |&alpha;|AVLips/AP|AVLips/AUC|SHDF/AP|SHDF/AUC|
> |---|---:|---:|---:|---:|
> |0|81.4|85.8|82.2|76.3|
> |learnable|83.0|86.2|84.1|78.0|
> |Ours|83.6|87.7|85.7|80.2|
>
> >**Q3:** The “smoothed max” video aggregation is undefined.
>
> **R:** Thank you for pointing out this issue. We have now explicitly defined the smoothed max operator. Let $s_t$ denote the frame-level score of the $t^{th}$ frame, and let $F$ denote the total number of frames in the video. The video-level score is computed by:
> $$
> s=\log\sum_{t=1}^{F}\exp(s_t).
> $$
> This is a smooth approximation of the max operator.
>
> >**Q4:** There is an inconsistency in the description regarding Hallo2 and Hallo3.
>
> **R:** Thank you for pointing out this inconsistency. This is a writing error in the manuscript. The generator that contributes 500 forged samples is Hallo2, not Hallo3. We will correct this mistake throughout the revision. We apologize for the confusion.
>
> >**Q5:** It is unclear whether the comparison reflects algorithmic improvement or simply lack of target-domain adaptation.
>
> **R:** Thank you for the comment. All baselines and our method are trained on talking videos only, with no singing-domain adaptation, and SHDF is used exclusively as an unseen target domain for evaluation. Under this design, Table 1 does not measure how well methods perform after target-domain adaptation. Instead, it measures how well they transfer from talking to singing.
>
> In Table 2, when both AVH-Align and T-AVFD are retrained on singing data, T-AVFD still performs better. This means the advantage of our method does not disappear once target-domain adaptation is allowed. In addition, Table 4 controls the synthesis paradigm by comparing talking and singing samples generated under the same generator. Under this controlled setting, baselines still degrade much more on singing, while T-AVFD remains more stable. This shows that the gap is tied to the talking-to-singing domain shift itself, rather than only to generator mismatch or lack of adaptation.
>
> Finally, the pretrained components are not unique to our method. Baselines such as AVH-Align and LipFD also rely on strong pretrained representations, yet they still perform substantially worse under the same cross-domain setting. Therefore, the improvement comes from the proposed algorithmic design, not simply from using a stronger pretrained backbone.

---

> > ### Author Rebuttal · Reviewer_bUiz · 2026-04-01
> >
> > Thanks to the authors for the rebuttal. My main concerns are adequately addressed, and I will keep my original score.

---

> > > ### Author Response · Authors · 2026-04-01
> > >
> > > Thank you very much for your thoughtful follow-up and for confirming that the main concerns have been resolved. We sincerely appreciate your time and careful consideration.

---

### Official Review · Reviewer_LeZ6 · 2026-03-09

**Soundness:** 3
**Presentation:** 3
**Significance:** 3
**Originality:** 3
**Overall Recommendation:** 4
**Confidence:** 3

**Summary:**

This paper expands the evaluation scenario of audio-visual deepfake detection from conventional talking to singing. The paper proposes the T-AVFD detection framework. This framework extracts two types of features in parallel: first, it utilizes Alpha-CLIP combined with multi-granularity text prompts to learn facial semantic authenticity patterns; second, it employs a pre-trained lip-reading model to extract traditional audio-visual alignment features. Subsequently, the model adaptively fuses these two feature streams through a differential weighting mechanism. The main highlights of this work lie in its zero-fake-data training paradigm and the shift in the detection dimension. The model is trained exclusively on real talking videos. By introducing a vision-language model, it shifts the detection basis from low-level pixel-level artifacts to high-level semantic authenticity.

**Compliance With Llm Reviewing Policy:**

Affirmed.

**Final Justification:**

This paper expands the evaluation scenario of audio-visual deepfake detection from conventional talking to singing. The paper proposes the T-AVFD detection framework. The author effectively addressed my questions regarding the experimental setup of the paper during the rebuttal session, so I recommend WA.

**Key Questions For Authors:**

See Weakness.

**Limitations:**

See Weakness.

**Strengths And Weaknesses:**

**Strengths:**
1. The writing is clear and the argumentation is easy to follow. The extensive experiments in both the main text and the appendix demonstrate a solid effort from the authors.
2. The introduction of the Singing Head DeepFake dataset is a good contribution. If open-sourced, it will provide useful data for the research community.
3. The use of Alpha-CLIP and text guidance is an effective design. It successfully shifts the detection focus from low-level pixel artifacts to high-level semantic anomalies, which contributes to the model's strong robustness.

**Weaknesses:**
1. To obtain the stable face semantic $f$, the method directly averages the Alpha-CLIP features across all video frames. Would this operation average out and erase crucial temporal inconsistency cues (which are common in deepfakes)?
2. The text prompts use descriptions like "expressive eyes" for positive samples and "dull eyes" for negative ones. This assumption appears too idealized. In real-world scenarios, singers often close their eyes or show tired and numb expressions during emotional performances. Is there a risk that these genuine samples might be misclassified as fake based on these prompts?
3. The paper lacks an evaluation of resource consumption. What is the time and space overhead (computational cost and memory usage) of this method during inference and training, especially when compared to other baselines?

---

> ### Author Rebuttal · Authors · 2026-03-30
>
> **We sincerely thank Reviewer LeZ6 for the valuable comments.** We greatly appreciate and are encouraged by the reviewer’s recognition of the clear presentation, solid experimental validation, benchmark contribution, and effectiveness of our method. Below, we respond point by point to the concerns raised and will make the corresponding revisions in the revised manuscript.
>
> >**Q1:** Would averaging Alpha-CLIP features over all frames erase important temporal inconsistency cues in deepfakes?
>
> **R:** Thank you for your valuable comment. We agree that temporal inconsistency is important in deepfake detection. However, the averaged Alpha-CLIP feature in our method is not intended to capture temporal inconsistency. Its role is to provide a stable facial semantic representation for learning generalized authentic facial patterns. The temporal information is modeled in parallel by the lip-reading branch, which is pre-trained to learn temporally consistent audio-visual representations. Therefore, averaging does not erase the temporal inconsistency cues. Instead, it stabilizes the facial semantic branch, while the audio-visual branch preserves the dynamic evidence. This is exactly why the two branches are designed to be complementary.
>
> >**Q2:** Could the idealized text prompts cause genuine singing samples with non-typical expressions to be misclassified as fake?
>
> **R:** Thank you for this comment. Importantly, the goal of our model is to learn generalized facial authenticity patterns, and the text prompts are used to facilitate this learning rather than serving as the key signal by themselves. This is also why the learned facial semantic remains effective from talking to singing. Although different emotional performances may vary across scenarios, the underlying authenticity patterns are still informative for distinguishing real and fake samples.
>
> This interpretation is also supported by the added learnability comparison below. Even with no learnable token or all learnable token, the model still achieves competitive performance. Our existing ablations are consistent with this interpretation. Table 5 shows clear degradation when removing face feature or FAPL. Therefore, the main gain comes from the learned facial authenticity patterns under the structured multi-granular formulation, rather than from the text prompts alone.
> |Learnability|SHDF/AP|SHDF/AUC|AVLips/AP|AVLips/AUC|
> |---|---:|---:|---:|---:|
> |no learnable token|83.8|77.6|82.1|86.2|
> |all learnable token|81.9|74.8|80.8|84.9|
> |Ours|85.7|80.2|83.6|87.7|
>
> To further verify this, we have added a prompt-source comparison. ChatGPT performs best, while Gemini and manual prompts also remain effective. This indicates that our model does not depend on specific idealized text prompts, although prompt quality and specificity can affect the final performance.
> |Generator|THB/AP|THB/AUC|SHDF/AP|SHDF/AUC|
> |---|---:|---:|---:|---:|
> |manual|83.3|88.9|81.9|74.1|
> |Gemini3|85.4|91.2|82.6|77.0|
> |ChatGPT4|87.6|93.0|85.7|80.2|
>
> The risk of misclassification of genuine samples cannot be completely excluded, since singing is inherently a more challenging audio-visual scenario. However, even under this harder domain shift, our method still consistently outperforms the baselines, indicating that the learned real facial semantic remains effective rather than simply overfitting to idealized prompt descriptions.
>
>
> >**Q3:** The paper lacks an evaluation of resource consumption. What is the time and space overhead of this method during inference and training, especially when compared to other baselines?
>
> **R:** Thank you for the comment. We note that the resource consumption of our method has already been reported in Appendix A.3.5.
>
> We agree that a comparison with baselines is valuable. To address this point, we have compared our method with AVAD and AVH-Align, which are also trained on real data only. It reports the training and inference cost on 3,000 samples. “–” means that the training code is not publicly available, so the corresponding cost cannot be reliably estimated. The comparison shows that our method achieves competitive training and inference efficiency, while delivering clearly stronger detection performance than the baselines reported in the paper.
> |Method|Training/Cost|Training/Time|Inference/Cost|Inference/Time|
> |---|---:|---:|---:|---:|
> |AVAD|-|-|~3.1GB|~80min|
> |AVH-Align|~3GB|~28min|~1.3GB|~1min|
> |Ours|~4GB|~36min|~1.7GB|~1.2min|
>
> >*Q:* If SHDF is open-sourced, it will provide useful data for the research community.
>
> *R:* Thank you for the positive comment. We will publicly release both the code and the SHDF dataset for the research community.

---

> > ### Author Rebuttal · Reviewer_LeZ6 · 2026-04-01
> >
> > Thanks to the authors. My concerns are addressed, and I will keep my original score..

---

> > > ### Author Response · Authors · 2026-04-01
> > >
> > > Thank you very much for your careful reading and for confirming that the concerns have been adequately addressed. We greatly appreciate your time and consideration.

---

### Official Review · Reviewer_si3f · 2026-03-10

**Soundness:** 3
**Presentation:** 3
**Significance:** 3
**Originality:** 3
**Overall Recommendation:** 4
**Confidence:** 2

**Summary:**

This work focuses on the generalization problem of cross-scenario audio-visual deepfake detection, especially the shift from conventional “talking head” videos to the more challenging “singing head” videos. Existing multimodal detectors rely heavily on lip-synchronization cues. However, in singing scenarios, rhythmic vocalization and musical accompaniment greatly weaken such cross-modal coupling, introducing non-negligible domain shift and leading to significant performance degradation of current detection models. To address this challenge, the authors first construct a new singing head deepfake dataset generated by advanced rhythm-aware synthesis models to fill the gap in this field. Second, the authors propose a text-guided audio-visual deepfake detection framework, which is trained only on real talking videos. It includes a facial authenticity pattern learner that leverages multi-granularity text prompts to extract generalized facial authenticity semantics. Meanwhile, a multimodal differential weight learning module is introduced to adaptively fuse these facial semantic patterns with conventional audio-visual alignment features. Experimental results demonstrate that the proposed method achieves strong cross-scenario generalization on various deepfake videos and shows certain robustness against multiple real-world perturbations.

**Compliance With Llm Reviewing Policy:**

Affirmed.

**Final Justification:**

Thanks for authors, I have no problems.

**Key Questions For Authors:**

1.Given the sensitivity of the modulation vector α shown in the ablation experiments, have you tried setting α as a learnable parameter so that it can dynamically adapt based on the latent features of the input video?
2.The text prompts used to guide learning are generated by ChatGPT and fixed. How sensitive is FAPL to the specific wording of these prompts? Would introducing an automated prompt learning or fine-tuning mechanism to adapt to different singing styles and facial conditions lead to more discriminative feature representations?
3.This method relies on two large pre-trained models: Alpha-CLIP for extracting visual semantics, and a lip-reading expert model for audio-visual alignment features. Although the authors mention in the appendix that inference is fast after feature caching, what is its computational overhead in practical deployment?

**Limitations:**

yes

**Strengths And Weaknesses:**

Strengths:
1.Extending the focus of audio-visual deepfake detection from the talking scenario to the singing scenario is a direction of great practical value. The facial dynamics during singing are more complex, and the pattern of lip synchronization changes, which presents a valuable new challenge for the deepfake detection community.
2.The proposal of the SHDF dataset fills the gap in benchmark evaluation for singing-oriented deepfake detection. The dataset is constructed rigorously using a variety of advanced rhythm-aware generation models, and strict user studies have been conducted to ensure the high fidelity of the forged videos.
3.This work reduces overfitting to alignment artifacts in specific datasets, and instead learns generalized facial authenticity patterns via vision-language foundation models. Through contrastive text-facial alignment, it provides strong prior knowledge for distinguishing real and fake samples. The strategy is intuitive and effective.
4.The proposed T-AVFD demonstrates strong cross-dataset generalization ability. When trained only on real talking videos, the model achieves promising performance not only on unseen singing videos but also on the diffusion-based THB dataset. In addition, it shows certain stability against real-world image degradations.

Weaknesses:
1.The modulation vectors used in the multimodal differential weight learning module are manually designed, and Table 10 clearly shows that model performance fluctuates significantly with these settings.
2.Although the SHDF dataset is of high quality and well-validated, its scale is still relatively small and its categories are quite limited compared with large-scale talking face datasets. This may limit its future use for training large detection models that heavily rely on massive data.
3.The positive and negative text pairs rely on manual definition, which may make it difficult to adapt to more complex facial expressions or scene variations across different singing styles and facial movements.

---

> ### Author Rebuttal · Authors · 2026-03-30
>
> **We sincerely thank Reviewer si3f for the constructive comments.** We greatly appreciate and are encouraged by the reviewer’s recognition of the practical value of our problem setting, the benchmark contribution, the effectiveness of our methodological design, and the strong experimental results. Below, we respond point by point to the concerns raised and will make the corresponding revisions in the revised manuscript.
>
> >**Q1/W1:** Have you tried setting &alpha; as a learnable parameter?
>
> **R:** Thank you for the question. &alpha; is manually specified, but it is not the fusion weight itself. It acts as a bias on top of the adaptive weights predicted by the weights generator. We have compared &alpha; = 0, learnable &alpha;, and our &alpha;. &alpha;=0 degrades performance, while learnable &alpha; improves on this setting but remains below ours. The sensitivity in Table 10 is expected, because different &alpha; settings shift the fusion bias among cues whose usefulness differs between talking and singing. The results indicate that the adopted &alpha; is not an arbitrary manual setting, but a manually specified and experimentally validated prior for cross-scenario transfer.
> |&alpha;|AVLips/AP|AVLips/AUC|SHDF/AP|SHDF/AUC|
> |---|---:|---:|---:|---:|
> |0|81.4|85.8|82.2|76.3|
> |learnable|83.0|86.2|84.1|78.0|
> |Ours|83.6|87.7|85.7|80.2|
>
> >**Q2/W3:** How sensitive is FAPL to the specific wording of ChatGPT-based prompts? Would introducing an automated prompt learning lead to more discriminative feature representations?
>
> **R:** Thank you for the question. To examine the sensitivity of FAPL to the specific wording of ChatGPT-based prompts, we have added the prompt comparison below. The results show that the full prompt performs best, followed by the face prompt and then the eye prompt, suggesting that FAPL benefits more from multi-granular complementarity than from any single wording choice.
> |Prompt|SHDF/AP|SHDF/AUC|THB/AP|THB/AUC|
> |---|---:|---:|---:|---:|
> |face|80.5|73.0|80.2|91.1|
> |eyes|77.6|68.9|78.1|89.6|
> |mouth|74.1|67.2|74.5|88.9|
> |full|85.7|80.2|87.6|93.0|
>
> Furthermore, we have added a comparison among prompts from different sources under the same multi-granular structure. ChatGPT performs best, while Gemini and manual prompts also remain effective. This indicates that FAPL does not rely on one specific wording only, although prompt quality and specificity still affect the final performance. Prior work [1] supports our prompt design based on the face, eyes, and mouth for deepfake detection. In addition, each prompt is augmented with learnable tokens, so the model does not rely on fixed text alone. *[1] Towards More General Video-based Deepfake Detection through Facial Component Guided Adaptation for Foundation Model, CVPR’25.*
> |Generator|THB/AP|THB/AUC|SHDF/AP|SHDF/AUC|
> |---|---:|---:|---:|---:|
> |manual|83.3|88.9|81.9|74.1|
> |Gemini3|85.4|91.2|82.6|77.0|
> |ChatGPT4|87.6|93.0|85.7|80.2|
>
> To examine how automated prompt learning affects performance, we have compared no learnable token, all learnable token, and our current design. Both alternatives underperform our method, suggesting that neither a fully fixed design nor fully automated prompt learning yields more discriminative representations in our setting. Instead, the best performance comes from combining predefined multi-granular prompts with learnable adaptation.
> |Learnability|SHDF/AP|SHDF/AUC|AVLips/AP|AVLips/AUC|
> |---|---:|---:|---:|---:|
> |no learnable token|83.8|77.6|82.1|86.2|
> |all learnable token|81.9|74.8|80.8|84.9|
> |Ours|85.7|80.2|83.6|87.7|
>
> >**Q3:** What is its computational overhead in practical deployment?
>
> **R:** Thank you for this comment. In practical deployment, the full uncached pipeline takes about 0.7 s per sample on an NVIDIA A100 GPU, with a peak memory usage of about 5 GB. These results indicate that the method has practical potential for real-world deployment.
>
>
> >**W2:** This scale and categories of SHDF may limit its future use for training large detection models that heavily rely on massive data.
>
> **R:** Thank you for this comment. We agree that, compared with large-scale talking face datasets, SHDF is still limited in scale and category coverage. However, SHDF is not designed as a large training corpus. In this work, its main role is to provide the first dedicated benchmark for the talking-to-singing setting, so that cross-scenario generalization can be evaluated explicitly. All baselines and our method are trained only on talking videos, while SHDF is used to test unseen singing generalization. Therefore, although SHDF may not yet support data-hungry large-model training, it still fills an important gap as a controlled benchmark for singing-oriented deepfake detection. We will discuss future extensions of SHDF in terms of larger scale and broader category coverage.

---

> > ### Author Rebuttal · Reviewer_si3f · 2026-04-01
> >
> > Thanks for authors, I will raise my score.

---

> > > ### Author Response · Authors · 2026-04-01
> > >
> > > Thank you very much for your careful reconsideration and for raising the score. We greatly appreciate your time and thoughtful evaluation.

---

### Official Review · Reviewer_o35L · 2026-03-11

**Soundness:** 3
**Presentation:** 3
**Significance:** 3
**Originality:** 3
**Overall Recommendation:** 4
**Confidence:** 4

**Summary:**

This paper pioneers the extension of audio-visual deepfake detection to the singing scenario, quantifying the talking-to-singing domain shift via MMD² and score distribution overlap, and constructing SHDF—the first dedicated singing head deepfake dataset. It proposes the T-AVFD framework, which models facial authenticity patterns to compensate for the weakened lip-audio alignment in singing and achieves robust cross-scenario detection through adaptive multi-modal fusion. The work fills a key research gap with innovative designs and solid experiments, yet has notable limitations in dataset construction and module design verification that need further optimization.

**Compliance With Llm Reviewing Policy:**

Affirmed.

**Final Justification:**

Most of the concerns are solved.

**Key Questions For Authors:**

1. When designing the FAPL module, the paper restricts the granularity of facial text descriptions to three fixed levels (face, eyes, mouth) and does not attempt to let the model autonomously mine the core semantic dimensions for real/fake discrimination. What fine-grained facial features do the authors consider to be the key for distinguishing real and fake samples in the singing scenario, and have any preliminary explorations or verifications been conducted?

2. Facial authenticity was not included in the evaluation dimensions when selecting generative models for the SHDF dataset’s fake samples. If the generative models are reselected based on the facial authenticity metric, how would it affect the challenge of the dataset and the detection performance of the T-AVFD framework on this dataset?

**Limitations:**

yes

**Strengths And Weaknesses:**

**Strengths:**

1. Precisely fills the research gap with quantitative domain shift analysis: It is the first to explore singing-scenario deepfake detection, quantifying the talking-to-singing domain shift with MMD² and score distribution overlap, and clearly defining the new detection challenges in this scenario.

2. Addresses scenario pain points with facial feature compensation: It accurately identifies the reduced reliability of lip-audio alignment cues in singing, and models generalized facial authenticity patterns to effectively make up for the deficiency of traditional alignment-based signals.

3. Refines multi-modal fusion for dynamic scenario adaptation: The proposed MMDWL module enables adaptive fusion of facial, visual and audio features, solving the failure of fixed-weight fusion under domain shift and improving the model’s adaptability to different scenarios.

**Weakness:**

1. The SHDF dataset only covers 80 identities for real samples and uses merely 3 generative models for fake samples. The limited coverage of identities and models makes it difficult to fully verify the model’s generalization capability.

2. When selecting generative models for fake sample synthesis, only visual quality, head motion and lip-sync accuracy are evaluated, while the core facial authenticity dimension is ignored, leading to a logical conflict with the paper’s research focus.

3. It solely relies on ChatGPT for generating text prompts, without comparing other generation methods (e.g., other LLMs, manual transcription) or conducting ablation experiments, resulting in insufficient empirical support for the reliability of the FAPL module.

4. The granularity of facial text descriptions is restricted to 3 fixed levels, without enabling the model to autonomously explore optimal semantic hierarchies or mine finer-grained features for real/fake discrimination, which limits the performance potential of the FAPL module.

5. Although comparative experiments on different parameter configurations were conducted for the modulation vector α in the MMDWL module, little specific experimental data were presented. Moreover, the vector is fully set manually throughout the study without exploring autonomous optimization methods, leading to insufficient verification of the rationality of weight design and a lack of flexibility.

---

> ### Author Rebuttal · Authors · 2026-03-30
>
> **We sincerely thank Reviewer o35L for the constructive comments and encouraging recognition of our work.** Below, we respond point by point to the concerns raised and will make the corresponding revisions in the revised manuscript.
>
> >**Q1/W4:** The paper uses three fixed text granularities. What are the key facial features in singing scenario, and what preliminary explorations have been conducted?
>
> **R:** Thank you for the comment. The prompt design is neither arbitrary nor fully fixed. Beyond the predefined hierarchy, the prompt is augmented with learnable tokens. Prior work [1] supports our prompt design based on the face, eyes, and mouth. The added facial-region comparison below shows that the face feature is most important in singing, followed by the eyes. Table 5 reports the single-text setting, while Table 11 compares progressively finer 2-pair, 3-pair, and 4-pair designs, where the 4-pair setting further includes skin. The 3-pair setting gives the best overall balance across talking and singing. *[1] Towards More General Video-based Deepfake Detection through Facial Component Guided Adaptation for Foundation Model, CVPR’25.*
> |Prompt|SHDF/AP|SHDF/AUC|THB/AP|THB/AUC|
> |---|---:|---:|---:|---:|
> |face|80.5|73.0|80.2|91.1|
> |eyes|77.6|68.9|78.1|89.6|
> |mouth|74.1|67.2|74.5|88.9|
>
> We have also compared no learnable token and all learnable token. Both variants yield inferior results to our design, showing that the best setting combines predefined multi-granular prompts with learnable adaptation.
> |Learnability|SHDF/AP|SHDF/AUC|AVLips/AP|AVLips/AUC|
> |---|---:|---:|---:|---:|
> |no learnable token|83.8|77.6|82.1|86.2|
> |all learnable token|81.9|74.8|80.8|84.9|
> |Ours|85.7|80.2|83.6|87.7|
>
> >**Q2/W2:** How would reselecting generative models by facial authenticity affect dataset difficulty and T-AVFD performance?
>
> **R:** Thank you for raising this. Facial authenticity is indeed a valuable criterion for dataset construction. However, this is different from the authentic facial patterns modeled in FAPL, which are learned from real samples to capture broader authentic facial semantics. We have added generator-side facial authenticity as an evaluation dimension in generator selection. The results show that MEMO achieves the highest facial authenticity. The MEMO-based singing results in Table 4 show that all methods degrade under this harder setting, indicating that higher facial authenticity makes detection more challenging. Importantly, T-AVFD remains the best performance. This is because the test samples, though more realistic, still differ from the authentic facial patterns learned from real samples.
> |Generator|Naturalness|Coherence|Temporal Consistency|
> |---|---:|---:|---:|
> |EchoMimic|3.3|3.6|4.0|
> |Hallo2|3.5|3.3|3.7|
> |DreamTalk|2.3|3.8|3.9|
> |MEMO|4.4|4.5|4.2|
>
> >**W1:** The coverage of identities and generators in SHDF limits its ability to fully evaluate generalization.
>
> **R:** Thank you for the comment. SHDF is intended not to fully evaluate generalization, but to serve as the first testbed for studying the cross-scenario domain shift. In Table 4, the generator is fixed and only the content changes from talking to singing. The baselines degrade sharply even under the current identity and generator coverage, showing that the talking-to-singing shift itself is already a substantial challenge, rather than an effect driven mainly by limited diversity.
>
> >**W3:** FAPL relies on ChatGPT-generated prompts.
>
> **R:** Thanks for the comment. We have added a comparison among prompts from different sources. ChatGPT performs best, while Gemini and manual prompts also remain effective. This indicates that FAPL does not rely on ChatGPT alone, although prompt quality and specificity still affect the final performance. Table 5 shows drops when removing text prompts or using a single text, while Table 11 shows that 3pos./3neg. provides the best balance across talking and singing. It suggests that the main gain comes from the multi-granular prompts, rather than from the prompt source.
> |Generator|THB/AP|THB/AUC|SHDF/AP|SHDF/AUC|
> |---|---:|---:|---:|---:|
> |manual|83.3|88.9|81.9|74.1|
> |Gemini3|85.4|91.2|82.6|77.0|
> |ChatGPT4|87.6|93.0|85.7|80.2|
>
> >**W5:** &alpha; lacks empirical validation of its rationale and flexibility.
>
> **R:** Thank you for the comment. &alpha; is not the fusion weight itself, but a modulation prior applied to the adaptive weights predicted by the weights generator. To address this concern, we have compared &alpha; = 0, learnable &alpha;, and our &alpha;. &alpha;=0 degrades performance, while learnable &alpha; improves on this setting but remains below ours. These results, together with the modulation factor study in Table 10, indicate that the adopted &alpha; is not an arbitrary manual choice, but an empirically supported prior for cross-scenario balance.
> |&alpha;|AVLips/AP|AVLips/AUC|SHDF/AP|SHDF/AUC|
> |---|---:|---:|---:|---:|
> |0|81.4|85.8|82.2|76.3|
> |learnable|83.0|86.2|84.1|78.0|
> |Ours|83.6|87.7|85.7|80.2|

---

> > ### Author Rebuttal · Reviewer_o35L · 2026-04-02
> >
> > Most of the concerns are solved, I will keep the original score.

---

> > > ### Author Response · Authors · 2026-04-02
> > >
> > > Thank you very much for your careful reading and for confirming that most of the concerns have been addressed. We truly appreciate your time and thoughtful feedback.

---

### Decision · Program_Chairs · 2026-04-30

**Decision:**

Accept (regular)

**Comment:**

The paper introduces a new benchmark and framework for detecting audio-visual deepfakes in the challenging singing domain. I find the extension of the detection from talking scenario to the singing scenario is highly practical. All reviewers agree that the problem is well motivated. The initial concerns of the reviewers are adequately addressed by the authors in the rebuttal. Following reviewers consensus, I recommend weak accept.